# Inflammation: A New Look at an Old Problem

**DOI:** 10.3390/ijms23094596

**Published:** 2022-04-21

**Authors:** Evgenii Gusev, Yulia Zhuravleva

**Affiliations:** Institute of Immunology and Physiology, Ural Branch of the Russian Academy of Sciences, 620049 Ekaterinburg, Russia; jazhur@mail.ru

**Keywords:** general pathological process, inflammation, systemic inflammation, cellular stress, tissue stress, evolution of inflammation, neurodegeneration, atherosclerosis, tumors

## Abstract

Pro-inflammatory stress is inherent in any cells that are subject to damage or threat of damage. It is defined by a number of universal components, including oxidative stress, cellular response to DNA damage, unfolded protein response to mitochondrial and endoplasmic reticulum stress, changes in autophagy, inflammasome formation, non-coding RNA response, formation of an inducible network of signaling pathways, and epigenetic changes. The presence of an inducible receptor and secretory phenotype in many cells is the cause of tissue pro-inflammatory stress. The key phenomenon determining the occurrence of a classical inflammatory focus is the microvascular inflammatory response (exudation, leukocyte migration to the alteration zone). This same reaction at the systemic level leads to the development of life-critical systemic inflammation. From this standpoint, we can characterize the common mechanisms of pathologies that differ in their clinical appearance. The division of inflammation into alternative variants has deep evolutionary roots. Evolutionary aspects of inflammation are also described in the review. The aim of the review is to provide theoretical arguments for the need for an up-to-date theory of the relationship between key human pathological processes based on the integrative role of the molecular mechanisms of cellular and tissue pro-inflammatory stress.

## 1. Introduction

Inflammation is a universal response of an organism to predominantly local tissue alterations of diverse nature. According to the canons of general pathology, inflammation is a typical complex (local and systemic) general pathological process which forms the basis of disease pathogenesis with a variety of inflammatory focus localizations and symptomatology [1,2]. Classical (canonical) inflammation is characterized by a stereotypic complex of vascular changes, which lead to edema followed by migration of leukocytes to the damaged area and formation of an inflammatory focus [3]. The presence of a focus of inflammation is a key attribute of different variants of classical inflammation and its distinguishing feature from non-classical variants of inflammation [4]. The primary function of the inflammatory focus is to isolate the damage factor, then eliminate it and subsequently regenerate or repair (sclerosis) the injured tissue. The most evident phenomena of inflammation—redness (rubor), swelling (tumor), fever (calor), pain (dolor), dysfunction (functio laese)—were already described by the ancient Roman physicians Celsus and Galen (first and second century AD) [5].

Meanwhile, recent advances in molecular biology and medicine have shown that the molecular mechanisms of inflammation and immune response, both at the cellular and organ-organismal levels, are much more widespread than previously thought. They not only underlie the pathogenesis of a very broad range of somatic diseases that previously were not classified as “classical inflammation”, but they are also an integral part of even some physiological processes. The inability of theoretical medicine to revise the traditional views of inflammation as a set of biologically and clinically different general pathological processes has determined, as we think, the crisis of modern general pathology and pathological physiology as the sciences that study general regularities in the pathogenesis of various diseases. The consequence of this crisis is that the clinical definitions have come to reflect fundamental regularities in the pathogenesis of quasi-inflammatory diseases. Moreover, the available body of data on inflammation has gone far beyond the classical notions of inflammation without adequate theoretical justification. This concerns—first of all—the conceptual models of syndromes, such as the concept of systemic inflammatory response syndrome [6], which had been popular before the latest version of sepsis (Sepsis-3, 2016) was adopted [7], or the concept of metabolic syndrome reflecting the notions of chronic systemic low-grade inflammation (ChSLGI) [8,9]. However, these approaches prioritize clinical problems and cannot provide a theoretical basis for describing general patterns of human pathology.

The aim of the review is to provide theoretical arguments for the need for an up-to-date theory of the relationship between key human pathological processes based on the integrative role of the molecular mechanisms of cellular and tissue pro-inflammatory stress.

## 2. Cellular Stress as a Functional Unit of Pro-Inflammatory Tissue Stress

### 2.1. General Characteristics of Cellular and Tissue Stress

Cellular stress (CS) is a typical cellular response to any form of macromolecular damage aimed at restoring cellular and tissue homeostasis [10]. Cellular stress includes universal mechanisms based on a phylogenetically conserved set of genes and their activation pathways, and mechanisms specific to individual cell types within a multicellular organism [11]. Given the above, we propose the following definition: “Cellular pro-inflammatory stress is a complex of interrelated universal and population specific cellular processes in response to the action of real and potential damage factors” [4].

An individual cell is a morphofunctional unit of the organism with an integral system of genome, transcriptome, proteome, and metabolome. Thus, CS is an elementary functional unit (subsystem) of a more integral process—tissue stress (TS)—which is a response of a certain tissue or organism as a whole to the impact of damaging factors of various nature and includes universal (primarily, immune) and tissue-specific mechanisms aimed at maintaining or restoring the already disturbed homeostasis. These reactions are usually associated with inflammation, as well as with immune response in infectious, autoimmune, and allergic processes. However, TS reactions manifest themselves not only in classical and non-classical forms of inflammation, but also in many physiological processes [4] which can no more be defined as inflammation, since this term is associated with pathology. As already noted, in contrast to the canonical inflammation, its non-classical variants are not attributed to the processes of the inflammation focus. In general, CS can be considered as an elementary, but integral, functional unit of various pathological processes, and TS can serve as a common pathogenetic platform for their development. Figure 1 shows the principal differences of the most universal TS variants depending on the intensity of the damaging factors at the body level.

A key pathogenetic phenomenon that separates different variants of inflammation-related general pathological processes is the exudative response of microvessels at the local (in classical inflammation) or systemic levels. The presence of a transition zone between classical and systemic inflammation allows timely prediction of the onset of critical conditions in patients (Figure 2).

### 2.2. Triggers of Cellular and Tissue Stress, and Response Regulation

The factors that initiate CS and TS can be categorized as follows:Any damage to macromolecules (in cells and extracellular matrix) that is recognizable by CS sensors [11].Potentially dangerous disturbances of key homeostasis parameters: acid–base balance, temperature, osmotic and hydrostatic pressure, changes in cytoplasmic and mitochondrial levels of calcium cations and other electrolytes, and decrease in cellular concentrations of ATP, oxygen (hypoxia), and some metabolites [4].Of special note are the lipotoxicity factors that act on mitochondria and other cellular structures. These include: excessive contents of saturated free fatty acids (FFA), diacylglycerol, and ceramides, as well as modified carnitine, non-esterified cholesterol, and some other hydrophobic molecules [4,12,13,14].Recognition of alarm signals from pathogens and damaged tissues by the pattern-recognizing receptors (PRRs) of cells directly associated with inflammation: immunocytes, epitheliocytes, connective tissue cells, and endotheliocytes [15]. PRR ligands are represented by conserved microbial structures—pathogen-associated molecular patterns (PAMPs)—and endogenous, damage-associated molecular patterns (DAMPs). Receiving signals via PAMPs and DAMPs, cells can rapidly enter into a state of stress prior to being damaged and realize their pro-inflammatory and immunocompetent functions. Particularly noteworthy among the PRRs are two families: toll-like receptors (TLRs) and intracellular NOD-like receptors (NLRs) [16].Antigen recognition by antibodies (with subsequent action of immune complexes on cells) and by T-cell receptors (TCR), leading to a strong activation effect on both the T-lymphocytes and the cells interacting with them, above all the antigen-presenting cells.The action of various activators of the complement, hemostasis, and kallikrein–kinin systems followed by the effect of the activation products of these systems on various cells.Excitotoxicity in the ‘narrow sense’ is the toxic effect of high doses of glutamate [17,18] and some other neurotransmitters and their catabolic products [19] on neurons; in the ‘broad sense’, it is the pathological hyperactivation of cells by various regulatory molecules, primarily pro-inflammatory cytokines such as TNF-α and IL-1β. The latter manifestation of excitotoxicity is most prominent in the cytokine storm syndrome [20], including severe COVID-19 [21,22]. The phenomenon of cytokine excitotoxicity makes the development of CS ‘contagious’.

Cellular and tissue responses to damage in themselves include biologically aggressive factors, such as reactive oxygen species (ROS) [23], while the extreme stressor functions can modify individual homeostasis parameters [24]. At the same time, TS manifestations that are inadequate in severity, time, and space can themselves be a driving factor in the pathogenesis of various diseases [25]. Consequently, the CS and TS development has to be strictly regulated. This applies both to thresholds for the development of CS and TS and to subsequent progressions to more pro-inflammatory and therefore more biologically aggressive stages of CS and TS. Thus, the development of almost all CS and TS processes is controlled by the principle of negative feedback, which determines their reversibility. In particular, a large group (~30) of scavenger receptors (SRs) plays a key role in the uptake by macrophages and some other cells of aberrant cells and metabolites, including oxidized low-density lipoproteins (oxLDL) and advanced glycationend-products (AGE), as well as various PAMPs and DAMPs [26]. These receptors function at the interface between immunity and metabolism—as well as between normality and pathology—and are involved in the regulation of CS and TS. In particular, they form receptor clusters with TLRs, tetraspanins, and other receptors and can multidirectionally model the passage of activation signals into the cell depending on the prevailing situation and SR types involved in cell activation [27,28,29,30].

### 2.3. Particular Typical Processes of Cellular Stress

In one capacity or another, CS is inherent in all types of cells, but primarily in the immune system, for which pro-inflammatory stress is a prerequisite for the performance of its main functions. In this case, the following universal and interrelated components of cellular stress can be distinguished (Figure 3) [4,31,32,33,34,35,36]:

Oxidative stress—Oxidative stress develops in a cell when the accumulation of pro-oxidants disturbs the redox equilibrium and causes an imbalance between the oxidants and antioxidants in favor of the oxidants [37,38]. The accumulation of ROS in the nucleus contributes to DNA damage, so the redox equilibrium in this cellular compartment is relatively stable. The main site of ROS formation under CS is the mitochondria [39]. In the cytoplasm, ROS generation occurs with the participation of cytochrome C released from the mitochondria as well as NADPH-oxidases of microsomal oxidation, 5-lipoxygenase, xanthine oxidase, and cytochrome P-450. They directly or indirectly activate many of the receptors, transcription factors (TFs), and protein kinases associated with CS development. Both excessive and insufficient development of oxidative stress in response to damage can be a trigger for the onset and progression of a wide range of human diseases [40,41,42].DNA-damage response (DDR)—Cells have developed the capacity for DDR to be able to control genotoxic stress and maintain accurate transmission of genetic information to subsequent generations. The accumulation of DNA damage in the cell leads to a number of alternative outcomes of DDR, including cell cycle arrest, senescence, malignization, or apoptosis [43]. In human cells, more than 1000 proteins are involved in the DDR process. These are primarily nuclear chaperones (ubiquitin, nucleophosmin and SUMO protein), nuclear protein kinases (ATM, ATR, DNA-PKcs and Chk1/2), various nucleases, polymerases, ligases and DNA glycosylases, and many TFs, especially p53. The main function of DDR is to stop the cell cycle to enable DNA repair and cell survival [35]. At the same time, the process of apoptosis is an extreme variant of DDR aimed at preventing malignization and making it impossible to transmit genetic abnormalities to daughter cells.Mitochondrial stress, including mitochondrial unfolded protein response (UPR^mt^)—Mitochondria are the main donors of ATP and ROS, and the end point of catabolism and the starting point for anabolism. There are approximately 1500 proteins functioning in human mitochondria, of which only 13 are encoded in mitochondrial DNA (mtDNA) [44]. These are mainly the most important proteins of the mitochondrial respiratory complexes. The mitochondrial proteome is tuned to the functional status of its cell and depends on the action of activating and damaging factors on the mitochondrion itself. The extreme connection between the mitochondria and the cell nucleus is a well-established phenomenon that occurs in response to mitochondrial dysfunction. Various injuries in the proteome and mtDNA, including the accumulation of denatured proteins in the mitochondria, cause UPR^mt^ development. UPR^mt^ involves multidirectional changes in the biosynthesis of various mitochondrial proteins (reduction in potentially toxic proteins); and increased production and transport into the mitochondria of chaperones capable of repairing damaged mitochondrial proteins [45]. Integrative mitochondrial stress is primarily associated with the activation of ATF4 (activating transcription factor 4) [46] and the production of heat shock proteins (HSPs) and many kinases that integrate mitochondria into the CS system. Mitochondrial stress is aimed at eliminating mitochondrial damage and dysfunction. However, under certain scenarios, this program complex may fail to perform effectively, because individual mechanisms of mitochondrial stress may themselves become involved in the vicious pathogenetic circle that is characteristic of many diseases [18,47].Stress of the endoplasmic reticulum (ER), including calcium-dependent mechanisms and UPR^ER^—The disruption of ER integrity or accumulation of misfolded proteins in these cellular compartments initiates ER stress, primarily in the form of UPR^ER^ [33]. The UPR^ER^ process aims to restore an altered ER homeostasis by pursuing the following main objectives: (1) suspension of the synthesis and excretion of secretory proteins from the cell; (2) increased transcription of chaperones and other proteins involved in protein folding and protein maturation; (3) induction of denatured protein degradation via the ER-associated degradation complex (ERAD) [48]. UPR^ER^ is mediated by three main transmembrane sensors: (1) the inositol-requiring enzyme 1 (IRE1), (2) the protein kinase PERK, and (3) the transcription factor ATF6. They are all preserved in an inactive state, mainly by virtue of the BiP/GRP78 chaperone coupled to them [49]. Under ER stress, this chaperone binds to and is blocked by various unfolded proteins and thereby releases UPR^ER^ inducers in the active state. Thus, signal transduction via PERK, IRE1, and ATF6 provides a coordinated response that contributes to overcoming the impaired ER proteostasis. Prolonged or intensity-critical UPR^ER^, in turn, induces apoptosis through several pathways, including excess Ca^2+^ release into the cytoplasm from the ER. However, because ER stress activates the anti-apoptotic pathways (anti-apoptotic proteins of the Bcl-2 family) as well, apoptosis is an extreme and far from the only variant of the ER stress outcome [33].Response of inducible HSPs, including their participation in the UPR [50]. The HSPs response is an evolutionarily ancient and highly conserved molecular response of the cell to disturbances in its protein homeostasis (proteostasis) [51]. HSPs are the main chaperones of UPR. In addition, they perform numerous regulatory functions that influence almost all major CS processes [52].Inhibition (during cell growth) or intensification of autophagy processes (utilization of altered organelles and macromolecules) and other manifestations of lysosomal stress—Autophagy is a catabolic process involving a lysosomal phase, which is conserved in the evolution of all eukaryotes and runs (or occurs) in all human cells. Autophagy is part of many physiological or pathological processes, and its severity increases significantly during starvation and severe CS [53]. In these cases, autophagy usually promotes cell survival. Normally, most damaged and short-lived proteins degrade by the proteasome pathway after they have been marked with ubiquitin. In CS, the ubiquitin–proteasome pathway is overloaded and could act as an additional autophagy activating mechanism [54]. Besides, many long-lived proteins, large protein aggregates, and individual organelles can only be utilized by the process of autophagy with the participation of lysosomes and numerous supporting protein factors. In particular, mitophagy is the only mechanism for the physiological recycling of mitochondria [55]. Thus, autophagy largely determines the balance between protein biosynthesis, organelle biogenesis, and organelle degradation. Moreover, autophagy can also be crucial in preventing cell apoptosis or necrosis by removing damaged and pathologically activated mitochondria as well as various protein complexes and intracellular parasites. Autophagy is broadly divided into three main types and has several levels of regulation by CS mechanisms [56]. As cells age, autophagy regulation and realization may be disbalanced [57]. In particular, in normal aging and neurodegenerative diseases, the balance between the number of mitochondria (directly dependent on mitophagy intensity) and their degree of dysfunction (also dependent on mitophagy, but in the opposite way) may be maintained or disturbed [58].Inflammasome formation—Inflammasome is a multimeric cytosolic protein complex with sensory molecules in the form of intracellular PRRs of two families: NLRs (mostly) or absent in melanoma 2-like receptors (ALRs). During protein complex assembly, these receptors bind to procaspase-1, after which procaspase-1 is converted to caspase-1. Caspase-1 then induces the processing of IL-1β and IL-18 and, under certain conditions, the development of pyroptosis (programmed necrosis) [59]. Inflammasome formation is a sign of a relatively pronounced pro-inflammatory stress of immunocytes, epitheliocytes, endotheliocytes, and some other cells. Several additional conditions are necessary for the formation of an inflammasome, such as the activation of pro-inflammatory signaling pathways associated with the transcription factor NF-κB, oxidative stress buildup, and a decrease in the K+ concentration of the cytoplasm. The NLRP3 inflammasome assembly process is activated by the greatest variety of factors—namely: PAMP, DAMP, ROS, lysosomal proteinases, cholesterol crystals, β-amyloid, uric acid (metabolic DAMP), calcium phosphates, many exogenous irritants (e.g., asbestos and silicon), mtDNA release from mitochondria into cytoplasm, and recognition of internal and external PRRs signals [60,61,62]. The biological role of inflammasome formation is to enhance the development of inflammation and the immune response through pyroptosis, IL-1β production, and other factors [63]. However, the disruption of restrictive control over inflammasome formation can cause severe complications, especially in genetically determined autoinflammatory diseases [64].Formation of stress, non-coding RNAs, microRNAs—MicroRNAs (miRNAs) are small non-coding RNAs that, like long non-coding RNAs, have the ability to modulate gene expression at the post-transcriptional level either by inhibiting matrix RNA (mRNA) translation or by stimulating mRNA degradation [65]. The involvement of both types of non-coding RNAs in the pathologies associated with the development of CS has now become evident [40,66]. Moreover, miRNAs can also regulate CS development through intercellular communication, through the effects of extracellular vesicles containing non-coding RNAs [67].Formation of stress granules – At the post-transcriptional stage, RNA-binding proteins (RBPs) are a key contributor to the stress-induced regulation of the destiny and function of various RNAs [68]. At the same time, the function of stress granules down to a few microns in size, consisting of RNA and protein, is not yet fully understood [69]. Additionally, CS can induce in cells the formation of gel-like structures, including ones involving amyloid and prion-like proteins [70]. The formation of these structures is dynamic; they condense or dissolve quickly and are therefore ideal for participating in urgent cellular adaptation to stresses.Formation of an intracellular network of cellular stress signaling pathways—At the cell level, stress development is mediated by complex programs of epigenetic control and intertwining of signaling pathways whose protein elements are continually undergoing multiple posttranslational modifications [71]. Along with that, various extracellular and intracellular stress signals can activate common collector-type protein kinases (e.g., MAPK, Akt, PI3K, PKC, ATM, ATR, AMPK, PKA, PKR, mTOR) and key universal cellular stress transcription factors (e.g., NF-κB, p53, AP-1, HIF, HSF, NRF2, ATF4) in different cells. The same signaling molecules can be activated in different ways and participate in differently directed processes. However, in general, the polyfunctional factors may feature certain functional preferences. Thus, the key role in the development of CS in hypoxia is attributed to HIF-1 (hypoxia-inducible factor-1) [72]; in HSP production, to HSF1 (heat shock factor 1) [73]; in oxidative stress, to NRF2 which triggers antioxidant production through a negative feedback mechanism [74]; while ATF4, as already noted, plays a determining role in UPR^mt^ development. The dynamic network of signaling pathways integrates the different elements of the CS into a single whole, including the receptor and secretory phenotype of pro-inflammatory cells (Figure 3).Formation of pro-inflammatory receptor and secretory cell phenotype—Almost all nucleated cells, when activated, express inducible receptors and secrete a spectrum of inflammatory mediators, including cytokines [4,75,76]. This fact determines the possibility of cytokine network formation in all possible variants of TS. Thus, the emergence of a pro-inflammatory phenotype in a large number of cells at once causes a network effect with TS development [77].

Thus, the CS includes a number of typical interconnected functional blocks which form an integral system of cellular response to the action of damaging factors. Certain context and unequal expression of these blocks, as well as more particular manifestations of CS which are typical for individual cell populations (especially in the immune system), determine the specificity and functional orientation of CS and TS.

### 2.4. Outcomes of Cellular Stress

The following processes may be referred to typical CS outcomes:Cell adaptation to a damaging factor. By virtue of their ability to respond to CS, cells can become resistant to the damaging factor and recover intracellular and tissue homeostasis. Having achieved partial adaptation to the prolonged action of damaging factors, cells can sustain their pro-inflammatory status, forming a state of tissue allostasis.Apoptosis is an essential component of various processes, including normal cell turnover, proper development and functioning of the immune system, hormone-dependent atrophy of unnecessary tissues, embryonic development, and death of damaged or malignant cells without pro-inflammatory response [78]. Meanwhile, a level of apoptosis that exceeds the regenerative capacity of the organ promotes tissue atrophy and, consequently, sclerosis of the parenchyma [79,80]. The process of apoptosis is induced by many signaling pathways which can be subdivided into ‘extrinsic’ and ‘intrinsic’ depending to varying degrees on caspase engagement. External proapoptotic signals are directed towards receptors for cytokines of the TNF family, which are involved in the activation of proapoptotic caspases. The main intrinsic pathway of apoptosis is the result of increased mitochondrial permeability and the release of pro-apoptotic molecules into the cytoplasm, primarily of cytochrome C (activates caspase 9 and then other pro-apoptotic caspases) [81]. The mitochondrial response, in turn, is triggered and controlled by pro-apoptotic and anti-apoptotic proteins of the Bcl-2 family, IAP, and many other factors [82,83]. Dead, fragmented cells produced by apoptosis are rapidly taken up by stromal macrophages for final degradation without significant DAMP formation. Caspases that are involved in apoptosis are divided into initiators (2, 8, 9, 10, 12) and effectors (3, 6, 7) [81,84]. The complexity of the mechanisms of apoptosis regulation [85] determines the fact that this process may manifest itself in the development of different variants and stages of CS.Programmed cell necrosis: necroptosis, pyroptosis, NETosis, parthanatos, autophagia, “cornification”, oxytosis, ferroptosis, secondary necrosis, oncosis, sarmoptosis, autosis, autolysis, paraptosis, and “mitotic crash” [4,86,87,88,89,90,91,92,93,94,95]. The numerous designations of this process reflect differences in the signaling pathways and in the biochemical and morphological features of the process, including the activation of necrosis-specific caspase types [86]. Thus, we can distinguish several variants of programmed necrosis, which are associated with high pro-inflammatory levels of CS and the formation of high concentrations of DAMP: *pyroptosis* (associated with inflammasome formation) [63]; *NETosis* (neutrophil extracellular traps formation), which was originally associated only with neutrophils, but was later discovered in other professional phagocytes [90,91,92]; *autophagic cell death* is a term widely used to describe cases of cell death accompanied by massive cytoplasmic vacuolization [93]; *necroptosis*, associated with the activation of receptor-interacting protein kinase 1 and 3 (RIPK1, RIPK3) and formation of a protein complex known as necrosome [94]; *secondary necrosis*, which occurs when apoptotic cells are not cleared in a timely manner and the process progresses to a “late apoptosis” phenomenon [95].Metaplasia is associated with the development of CS and TS, e.g., in the metaplasia of airway epithelium [96], endometrium [97], or connective tissue [98]. Gastric epithelium metaplasia occurs against the background of inflammation and atrophic changes (especially of the glands) of the gastric mucosa [99]. The process of metaplasia tends to be progressive and presents a risk of malignization of the relevant tissue [99].Cell malignancy and malignant tumor formation are associated with failure of the DDR mechanisms, retention of multiple mutations useful for tumor cell survival but harmful to the organism, and formation of a ‘parasitic’ genome in tumor cells.Cell aging is caused by the stochastic accumulation of damage in biomolecules (in the genome, transcriptome, proteome) that are vital for proper cellular function. These changes provoke CS with ROS accumulation, cell cycle blockade, and cellular and tissue allostasis formation [100,101]. The aging process affects all cell types, including stem cells [102]. Cell aging entails a state of irreversible arrest of proliferation in which cells remain metabolically active and secrete a number of pro-inflammatory and proteolytic factors and other components of the senescence-associated secretory phenotype (SASP) [103]. Cell aging is characterized by morphological transformations, namely: high level of β-galactosidase (SA-β-gal) expression, accumulation of cyclin-dependent kinase 2A inhibitor protein p16INK4a, SASP, formation of heterochromatin foci (SAHF), accumulation of aberrant protein aggregates and granules in cells, telomere shortening, and oxidative stress [43,104,105]. Cell aging is an alternative (to malignancy) means of CS development, in which the cell continues to accumulate sublethal damage, which may finally lead it to some variant of cell death or persistent dysfunction. SASP includes growth factors, cytokines, and extracellular proteases that modulate most of the both beneficial and detrimental microenvironment phenotypes caused by ageing cells [106]. In this case, cell aging and pro-inflammatory SASP may form a vicious pathogenetic circle involved in the formation of aging tissue allostasis [107].

### 2.5. Stages of Cellular and Tissue Stress

The stages of CS are determined by a number of parameters, including the presence or arrest of the cell cycle, the characteristics of the cell’s pro-inflammatory phenotype; the ratio of anabolism and catabolism processes; the degree of insulin resistance; the differentiation features of parenchymal, connective tissue, and immune system cells; and the resistance of cells to damage, apoptosis, and programmed necrosis. All this makes it necessary to concretize the verifying features of CS stages in individual cell populations. Meanwhile, it is possible to broadly identify the universal features of the three stages of the CS (Table 1) as follows:

#### 2.5.1. Stage 1

The prevailing growth of anabolism over catabolism with increased tolerance to the action of damaging factors. This response leads to increased cell survival under extreme conditions and, at the same time, elimination of irreversibly damaged cells, accumulation of functional reserves, and enhanced cell adaptation to potential damage. At the tissue level, this stage manifests itself in the following processes: growth of the organism under physiological conditions, tissue response to relatively short-term exposure to low-intensity damaging factors, and the repair stage of the inflammatory process [108]. At the same time, the mechanisms of this stage may also perform pathologically, e.g., with the progression of tumor growth [109,110]. The molecules secreted at this stage will be dominated by growth factors and factors that limit inflammatory cell transformation, such as adenosine acting through purinergic P1 receptors (A1, A2A, A2B, and A3) [111,112]. Many of these factors act through G-proteins and the insulin-dependent class I PI3K/Akt2/mTOR pathway associated with growth and anabolism [109]. Further on, the process involves the glucose transporter type 4 (GLUT-4) and the transcription factor FOXO1 (fork head box protein O1), which determine the metabolic effects of insulin in facultatively glycosylating tissues [113,114]. The processes of anabolism are related to catabolism, which is a donor of energy and key metabolites needed for biosynthesis. In this case, ATP shortage initiates the activation of AMP-activated protein kinase (AMPK), which leads to the activation of lipolysis, proteolysis, autophagy, but also to the enhancement of glucose transport into cells through the involvement of GLUT-4 [113]. Therefore, the metabolic effects of AMPK in cells are essential for the implementation of the various stages of TS. However, under conditions of nutrient deficiency, AMPK hyperactivation acts as a metabolic limiter that inhibits cell growth, including by inhibiting mTORC1 and abolishing the inhibitory effect of mTORC1 on autophagy [115]. However, these effects are more characteristic of the later stages of the CS.

#### 2.5.2. Stage 2

This is a transitional stage to a more pronounced pro-inflammatory phenotype. At this stage, there is already a functional shift in favor of more pro-inflammatory forms of mitogen-activated protein kinases (MAPKs) and TFs [116]. Thus, the p53 and NF-κB signaling pathways may competitively inhibit each other [117]. For example, at relatively moderate levels of oxidative stress, NF-κB is not activated, but one can observe p53-mediated DNA repair or apoptosis of irreversibly damaged proliferating cells. A further increase in oxidative stress activates NF-κB and inhibits p53-induced cell apoptosis; this contributes to cell resistance to oxidative stress and enhances pro-inflammatory activity [118]. A general increase in ROS production in endotheliocytes may inhibit the constitutive NO synthase (cNOS), which at the systemic level may be one of the mechanisms of hypertension [119]. The pro-inflammatory phenotype at this stage will also be characterized by an increased role of pro-inflammatory cytokines, inflammasome formation, and involvement of purinergic P2 receptors (the main ligand being ATP). In particular, the P2X receptor in the presence of inflammasomes is a key mechanism for the development of pyroptosis [120]. This stage is characterized by insulin resistance, autophagy, and UPR enhancement being one of the conditions for cell survival.

#### 2.5.3. Stage 3

At this stage, pro-inflammatory changes and the cellular phenotype as a whole become more distinctly pathological, along with an increasing probability of cellular necrosis variants—such as pyroptosis, necrobiosis, and NETosis (in phagocytes)—as well as cell aging with progressive functional disturbances. These abnormalities are manifested by increased expression of NF-κB and the most pro-inflammatory forms of MAPKs [121]. As the pro-inflammatory phenotype progresses, the probability of inducible NO synthase (iNOS) expression in endotheliocytes and inflammatory macrophages increases [122]. At the tissue level, the prevalence of this stage in CS will promote atrophy and sclerosis, in particular the replacement of parenchymatous cells by more stress-resistant connective tissue elements (Figure 4). At the same time, the accumulation of necrotic cells in the organs may contribute to the transformation of a local low-grade inflammation into a classic type of inflammation [123].

A more detailed characterization of CS stages requires assessing the inflammatory phenotype in specific cell populations and subpopulations. At the same time, different cell types do not have the same resistance to the damaging factors of CS. Therefore, during chronic tissue stress, parenchymatous cells can be replaced by elements of connective tissue (tissue sclerosis process).

In reality, CS manifestations may be less clear and display mixed signs of different stages in individual cells. Moreover, the signs of these stages are undoubtedly more numerous than those presented in Table 1. At the organ and organism levels, the situation is even more complex since TS integrates different cell types that are not in the same ‘inflammatory status’. Meanwhile, studying processes both in vitro and, especially, in vivo requires a systematic characterization of the study object, which—we think—cannot be fully achieved without such generalizations and simplifications.

### 2.6. The Physiological Role of Cellular and Tissue Stress

Latent effects of tissue alteration at subthreshold levels for the development of inflammation may be companions not only to pathology, but also to many physiological processes. This is evidenced not only by the widespread occurrence of apoptosis in the organism [124], but also by the detection of certain cellular necrosis indicators in healthy individuals, including myoglobin, aminotransferases, and many other markers of tissue damage [125]. As was already noted, moderate manifestations of CS are associated with tissue regeneration as well as with embryogenesis. In particular, the Hippo signaling pathway (which controls organ size through the regulation of cell proliferation and apoptosis) is associated with various inflammatory modulators such as FoxO1/3, TNFα, IL-6, COX2, HIF-1α, AP-1, JAK, and STAT [126]. The liver is not only an organ of acute-phase response to inflammation; normally, it has actively functioning stromal macrophages (Kupffer cells) [127]. Moreover, hepatocytes involving cytochrome P450 and other oxidative stress mechanisms may be participating in the metabolism and utilization of xenobiotics [128]. Furthermore, TS may vary widely in the extent of its manifestations in working skeletal muscles, including oxidative stress, increased autophagy, hyperproduction of HSPs, and pro-inflammatory secretory phenotype [39,129,130]. These effects may significantly raise the levels of pro-inflammatory cytokines, especially IL-6, in the blood of athletes during competition [131]. However, the formation of NLRP3 inflammasomes in muscles is already a sign of pathology, such as sarcopenia [132]. The myocardium is more resistant to the development of CS, but experiments on rats show that CS and TS may develop under increased physical activity in this tissue as well [47].

In healthy integumentary tissues, pro-inflammatory TS may be of a stable nature, which can be termed as normal pro-inflammatory tissue tone. Thus, studies reveal that NLRP6 inflammasomes—but not the more pro-inflammatory NLRP3—already appear under physiological conditions in the mature cells of the intestinal epithelium, which contributes to an adequate interaction between the intestinal microflora, epithelium, and immune system [133]. NLRP6 deficiency in enterocytes may lead to infectious complications as well as polyp formation and cancer [133]. Pro-inflammatory tissue tone is also essential for the maintenance of the normal functions of the epidermis [134,135]. Furthermore, the presence of a pro-inflammatory tone can be seen in lymphoid organs not only because of the constant contact of immunocytes with antigens, but also because of potential autoantigens in lymphocyte selection in primary lymphoid organs [136].

Thus, tissues with a high pro-inflammatory tone are characterized by continuing contact with damaging factors, a close relationship with the immune system, a high degree of cell turnover, and relative resistance of cells to CS and TS factors. At the same time, it is possible to identify tissues sensitive to damage and pathogenic factors of TS (Figure 5). These tissues are characterized by high functional specialization, low regenerative capacity (except for the testes), low expression of pro-inflammatory phenotype, isolation from alteration factors, and the immune system by histohematic barriers; isolation of the vascular endothelium is ensured by glycocalyx [137]. It is these tissues, along with the liver and muscle, that are central to the pathogenesis of allostasis-related diseases, the chronicity of TS, and the development of ChSLGI.

## 3. Tissue Stress as a Common Pathogenetic Platform for Modeling Basic General Pathological Processes in Humans

It has now become obvious that different pathological processes have common, universal mechanisms of pathogenesis both at the cellular and organ-organismal levels (Figure 6). Moreover, as noted above, many of the mechanisms of inflammation (but not the process of inflammation as a holistic phenomenon) are also involved in physiological processes. Thus, despite the obvious differences, all these processes are interrelated and there is a need to allow for this relationship, including when prescribing pathogenetic therapy. Once again, it should be made clear that tissue pro-inflammatory stress is a broader con-cept than inflammation. Thus, CS and TS are the basis of any form of inflammation. However, physiological manifestations of pro-inflammatory tissue stress cannot be con-sidered as inflammation, because inflammation is an a priori pathological process. It is now well known that the causes of tumor growth are often associated with chronic in-flammation in various organs [138,139]. At the same time, perifocal inflammation in the tumor growth area is one of the immune system’s responses to tumor expansion [140,141]. Although tumor growth as a typical pathological process is not inflammation itself, it is nevertheless closely linked to the mechanisms of pro-inflammatory cellular/tissue stress (Section 3.3). In general, tumor growth is both the result and inducer of tissue damage, but unlike inflammation it is not a form of a genetically determined body response to damage. The same is true for the processes of accelerated tissue aging, tissue atrophy, and a number of other pathological processes which have tissue pro-inflammatory stress mechanisms at their core and can be associated with inflammation, but are not identical with inflammation as a general pathological process.

As shown in Figure 6, all basic pathological processes may be divided into three basic blocks: (1) classical forms of inflammation; (2) tumor growth; (3) non-classical quasi-inflammatory processes. These, in turn, can be divided into high-intensity systemic inflammation and low-intensity local and systemic inflammation (more correctly, para-inflammation). In this context, atherosclerosis as a specific form of a general pathological process will have separate signs of both productive classical inflammation and local para-inflammation. It should be stressed that systemic inflammatory response (SIR) is not an independent form of general pathological process, since SIR is a sign of ChSLGI, systemic changes in classical inflammation, and microcirculatory disorders in high-intensity systemic inflammation. Thus, SIR is a symptom of various general pathological processes and, therefore, this phenomenon needs quantitative and qualitative characterization and comparison with other parameters of the studied processes.

### 3.1. Tissue Stress Variants in the Focus of a Classical Inflammation

The main causes of classical inflammation are strong local immune response to infection, autoantigens, allergens, and tissue necrosis (Figure 6), as well as genetically determined mechanisms of autoinflammatory diseases [142]. In these cases, TS is characterized by microvascular reaction and migration of cellular and humoral factors of the immune system to the damage area, that is, it leads to the formation of an inflammation focus. The focus of inflammation, as already noted, is an attribute of classical inflammation. Its main functions are isolation and elimination of damaging factors, as well as initiation of regenerative processes in the damaged tissues.

The main cause of inflammation focus formation is the intensity of damaging factors’ action above the threshold for pro-inflammatory microvascular reaction (Figure 2). However, in chronic pathologies, this process can develop long-term in the form of gradual trans-formation of low-grade inflammation into classical inflammation. For example, the characteristic signs of an inflammatory focus may emerge with progression of a local low-grade inflammation—such as in the retina [143], in non-alcoholic fatty liver disease [144,145], and in diabetic kidney disease [123]. In some cases, pro-inflammatory reactions of classical inflammation are not confined to the focus. Thus, systemic reactions are aimed at ensuring the functional status of the foci manifest themselves as a stress reaction of the neuroendocrine system, fever, recruitment of leucocytes from the bone marrow, and increased synthesis of acute-phase proteins in hepatocytes [146,147,148,149]. However, these reactions, as noted above, must be separated from systemic inflammation as a specific type of general pathological process. The life-critical microcirculatory disorders are the pathogenetic basis of systemic inflammation

The focal point of classical inflammation is the purulent destruction of tissue infected with extracellular pathogens, where IgG, C-reactive protein (CRP), other acute-phase proteins and neutrophils hyperactivated to NETosis are the main players, along with vascular reaction and complement, kallikrein–kinin, and hemostasis systems [150,151,152]. However, productive (proliferative-cellular) inflammation demonstrates the greatest variety of manifestations, and its main players are T lymphocytes, inflammatory macrophages (M), and, in some variants of inflammation, granulocytes of the cellular infiltrate.

During the development of a productive inflammation, macrophages undergo morphofunctional differentiation and are polarized in two main competitive pathways: the classical type of activation and differentiation in M1, and the alternative type in M2 [150]. These types of macrophages interact cooperatively with lymphocytes, primarily with various types of CD4 T-helper cells (Th-1, 2, 17, Treg). The current classification of M formed from monocytes under various in vitro stimuli is not limited to two types and includes at least 10 subpopulations in the M1–M2 range [153]. This differentiation is probably even more complex in vivo [154]. Additionally, Th differentiation is also characterized by plasticity. Thus, certain spectra of cytokines may bring about transformations as follows: Treg to Th17 or Th2, Th17 to Th1 or T cells with a plastic phenotype, and Th2 to CD4+ T cells, which can simultaneously produce cytokines of competitive Th types—namely, IL-4 and IFNγ [155,156,157]. In general, the Th1, Th17, and Th2 subpopulations—such as M1 and M2—are heterogeneous and can be subdivided into more private subpopulations [158,159]. In a simplified form, inflammatory macrophages can be divided into four subsets, each of which collaborates with Th subpopulations that are complementary to them and, thus, form four principal immune response vectors (I) shown in Table 2. Such subdivision is conditional. Rather, we can speak of certain corridors within which morphofunctional changes of immunocompetent cells (i1, i2, i3, i-reg) occur. The boundaries of these corridors are determined by the nature of immune response triggers, the influence of genetic and associated environmental factors, specific features of the cytokine network, and other mechanisms of extracellular communication, including extracellular vesicle exchange [160,161]. Often, even competing immune responses have mutual overlap zones. In particular, progressive interstitial renal fibrosis may result from complex mechanisms that arise from the interaction of M1 and M2 macrophages [162].

Thus, the immune response of T-lymphocytes to antigenic stimuli is closely linked to the development of inflammation. Antigen stimulation of immunocytes in lymphoid organs leads to the activation of their signaling pathways, including both TFs universal for cellular stress (e.g., NF-κB) and TFs that are responsible for vector T-cell differentiation (T-bet, FOXP3, families: STAT, GATA, ROR, and others). Next, highly differentiated Th subpopulations, together with innate immune cells, are involved in the formation of a cytokine network and a specific variant of productive inflammation in the inflammation focus. Th and other ‘inflammatory’ cells in the focus differentially secrete chemokines of the CXCL and CCL families, thereby attracting migrating cells that correspond to the immune response vectors formed in the focus (Table 2). The infectious agents of the inflammation focus, in turn, seek to actively disrupt the differentiation and viability of the immunocytes and deform the cytokine network of the tissue stress of the focus to their advantage [22,172].

### 3.2. Typical Patterns of Autoimmune Pathologies Can Be Considered as a Special Form of General Pathological Inflammatory Process

The body’s own tissue damage may be induced by the mechanism of autoinflammatory disease as a result of uncontrolled antigen-specific activation of innate immune factors [173] or by autoimmune response as a result of the effect of autoantigen-specific T-cells and antibodies [174]. When the autoimmune response leads to tissue alteration and autoimmune inflammation, we can talk about the development of an autoimmune pathological process. At the same time, the autoimmune process usually runs in a chronic progressive manner because the immune system wrongly recognizes its own antigens as potentially damaging factors, but cannot eliminate them from the body for obvious reasons. Typically, the autoimmune process manifests itself as a productive (with the predominance of i1 and i3 vectors), fibrinous, or mixed variant of classical inflammation, but can be complicated by microvascular disorders according to the variant of chronic systemic inflammation [123,175,176,177].

The immune system has multi-stage barriers to the initiation and development of autoimmune processes. These protective mechanisms include negative selection of lymphocytes, shielding of potential autoantigens, the presence of immunoprivileged organs, the presence of immunosuppressive components in the immune response, and other mechanisms [178]. However, all these barriers can be overcome where there is genetic predisposition to the impact of various ontogenetic and environmental factors that are autoimmunity triggers [179,180]. Of major importance are the following causes of autoimmune aggression, which can have a role to play in various infections as well [181,182,183]:Molecular mimicry of microbial proteins.Bystander activation—the release of autoantigens from tissue damaged by inflammation.Breakdown of biological barriers in immunoprivileged organs (central nervous system, eyes, testes, placenta), opening access to potential autoantigens for adaptive immunity.Polyclonal activation of lymphocytes in response to microbial super-antigens, or other factors activating potentially autoreactive T- and B-lymphocyte clones.Epitope spreading, a situation where autoimmune response targets do not remain the same but can diversify to include other epitopes on the same protein or on other proteins in the same tissue.Deficiency of the immunosuppressor vector i-reg in the processes of immune inflammation development.

It is important to remember that not every process of polyclonal lymphocyte activation leads to an autoimmune response, and not every autoimmune response results in tissue damage and the development of an autoimmune pro-inflammatory process, and the latter does not always lead to the development of a formal (canonical) autoimmune disease. Moreover, an autoimmune response may also develop in physiological conditions [184]. There are two main variants of the autoimmune process that can be identified in pathology, such as the development of canonical autoimmune diseases with the dominant role of autoimmune mechanisms in their pathogenesis, and latent manifestations of autoimmune mechanisms as additional, non-main factors of pathogenesis in infectious and other formally non-autoimmune variants of inflammation. Both of these variants, in particular, can occur as components in the pathogenesis of COVID-19 infection or its complications [22,185].

Thus, the autoimmune process has its own specific features, being, at the same time, closely related to other variants of inflammation and having common typical mechanisms of tissue pro-inflammatory stress with them. In pathogenetic terms, the typical manifestations of autoimmune diseases can be integrated into the overall system of the theory of general pathological processes.

### 3.3. Tumor Tissue Is under Tissue Pro-Inflammatory Stress

As noted above, tumor growth is associated with inflammation, but is not directly a form of inflammation. Meanwhile, recent molecular studies show a direct link between the presence of tumor tissue in the organism and specific mechanisms of cellular and tis-sue pro-inflammatory stress [38,186,187]. Malignant neoplasms form a parasitic system (essentially, an anti-system of the organism), including the tumor cells as such, the vascular network, tumor-associated macrophages (TAMs), and immune system cells that migrate to the tumor tissue [188,189]. Tissue structures of the tumor microenvironment on the one hand are a necessary condition for tumor growth, and, on the other hand, are a platform for the development of inflammarion and other mechanisms of antitumor response [190,191]. Anti-tumor immunity factors, hypoxia, tumor cell genome, as well as the effects of anti-tumor therapy act as tumor alteration factors that initiate tissue stress in tumor tissue [192,193,194,195]. A tumor invasion, in turn, is an alteration factor in relation to the host organism and thus can induce the development of chronic inflammation [196]. In contrast, low-grade inflammation can be a risk factor for tumorigenesis [197].

Tumor cells and TAMs produce a large number of cytokines, including various growth factors, chemokines, and immunosuppressive cytokines [198,199,200], which can lead to significant increases in the cytokine concentration in blood plasma [201]. Pro-inflammatory and anti-inflammatory local and systemic chronic reactions of tumor tissue through epigenetic changes can promote tumor growth and invasion and may well be characterized in the terms of a special variant of pro-inflammatory TS which is in interaction with para-inflammation and, in some cases, with canonical inflammation in the surrounding tissues [196].

Stress programs in tumor cells are based on a lot of universal signaling pathways, including the omnipresent activation of TGF-β1 and TNF signaling, as well as the activation of key TFs (NF-κB, STAT, HIF, AP-1, p53, STAT), and protein kinases (mTOR, MAPK, PI3K, AMPK) [110,202,203,204,205,206,207].

The response to DNA damage (RDD) mechanisms that enable tumor cells to proliferate under conditions of massive mutations are of particular importance for cellular stress in tumor tissue [208,209,210]. Important mechanisms of this adaptation are mutations and changes in the functions of the transcription factor p53, which is key for the regulation of the cell cycle and RDD [211,212]. As in other cells, pro-inflammatory status in tumor cells is associated with oxidative stress [37,42,213,214] and UPR development during mitochondrial and ER stress [215,216,217]. In some cases, the formation of inflammasomes in the tumor cells and in their microenvironment also contributes to tumor growth [218,219]. Furthermore, experimental evidence strongly suggests that regulatory non-coding RNAs function either as tumor suppressors or as oncogenes which are involved in the regulation of one or more cancer hallmarks, including evasion of tumor cell death, and their expression is often altered during cancer progression [220].

Thus, in our opinion, the universal features of tumor growth should be considered within the framework of a fundamental model of the general pathological process that is interlinked with other pro-inflammatory processes, that have a common pathogenetic basis with tumor growth in the form of tissue pro-inflammatory stress.

### 3.4. Chronic Systemic Low-Grade Inflammation

In our opinion, the general patterns of chronic low-grade inflammation (ChLGI), or para-inflammation, including ChSLGI, are as follows [4,123]:ChLGI is a manifestation of tissue stress in response to local or systemic damage at sub-threshold levels for the development of classical and systemic inflammation, respectively.The key triggers of ChSLGI are metabolic factors including: modified proteins (denatured, oxidized, glycated), high concentrations of saturated FFA and oxidized low-density lipoproteins (oxLDL), homocysteine, and many other metabolites. The progressive accumulation of genome, proteome, and metabolome injuries during aging contributes to the body’s pro-inflammatory status and the development of ChSLGI. Of particular importance in the development of ChSLGI are scavenger receptors of stromal macrophages, endotheliocytes, and some other cells, with these receptors being associated with metabolism, immunity, and inflammation [26].ChSLGI is characterized by moderate manifestations of SIR, namely: the elevation of C-reactive protein in the blood is usually in the borderline range of 3–10 mg/mL (a criterion for metabolic syndrome), and the elevation of pro-inflammatory cytokines is usually no more than 2–4 times over the upper normal range; the signs of significant tissue decay and systemic coagulopathy are not characteristic; the signs of organ dysfunction develop slowly as part of allostasis; and there is no direct association of these changes with systemic manifestations of infections and autoimmune diseases, i.e., with systemic manifestations of classical inflammation [221,222].The differentiation of local ChLGI from ChSLGI makes sense in the presence of a clinical presentation of these local abnormalities, for example in diabetic kidney disease [123].ChLGI involves a large number of parenchymatous and stromal cells of various organs, with relatively little involvement of inflammatory ‘professional cells’ (leukocytes and their progeny characteristic of the inflammatory focus). Therefore, ChLGI has no barrier function and no visible signs of classical inflammation.A key and integrating pathogenetic phenomenon of ChSLGI is endotheliosis, more specifically the pathological activation and dysfunction of endotheliocytes with the disruption of endothelial glycocalyx integrity in different parts of the vascular network [223].In ChSLGI, interrelated changes occur in key facultatively glycolating tissues (fat, muscle, liver), which leads to the development of insulin resistance and additional disturbance of metabolic homeostasis [57,224,225,226,227]. Therefore, the clinical presentation of ChSLGI is associated with morbid obesity, metabolic syndrome, sarcopenia, and type 2 diabetes mellitus. At the same time, the role of cellular and tissue aging is evident in the pathogenesis of these pathologies [227,228]. Moreover, atherosclerosis, osteoarthritis, neurodegeneration, hypertension, and chronic heart failure are typical local phenomena in aging and ChSLGI [229,230,231].

In general, tissues involved in ChSLGI are primarily those that have dynamic changes in CS and TS parameters under normal conditions as well (liver and skeletal muscle tissue), as well as tissues that are sensitive to damage and TS development; namely, vascular endothelium, brain, endocrine organs, myocardium, kidneys, and articular cartilage (Figure 7). In some cases, local ChLGI may transform into a classical type of inflammation, which further increases tissue destruction and the severity of internal organ sclerosis, for example, in the progression of non-alcoholic fatty liver disease (initially one of the clinical variants of hepatosis) [232]. Some local processes associated with ChLGI can be regarded as quite independent forms of general pathological processes, including age-related neurodegeneration and atherosclerosis.

### 3.5. Para-Inflammatory Neurodegeneration

The human brain is a complex system with different structures and cell types [233]. Despite the diversity of neurodegenerative diseases, their typical patterns can be identified, associated first of all with aging, including: genome instability, telomere shortening, DNA methylation and acetylation, other epigenetic changes, mitochondrial dysfunction, cellular stress with marked proteostasis disorders, as well as their interaction with pro-inflammatory tissue stress and associated proteinopathies in many brain regions [234].

The most common neurodegenerative proteinopathies causing proteotoxic stress are amyloidoses, tauopathies, α-synucleinopathies, and transactivation response DNA binding protein 43 (TDP-43) proteinopathies [235]. Abnormal protein conformers can spread between anatomical patterns, and their neuroanatomical distribution determines the clinical picture of specific diseases [235]. Changes in proteostasis are also associated with the specific features of cell types and stages of neurogenesis [236]. A common feature of neurons is the persistently high level of biosynthesis of many proteins, which limits the regulatory function of UPR and ambiguates the role of HSPs in the formation of insoluble protein complexes as metabolic alteration factors [237,238,239,240].

The other features of the central nervous system that determine the sensitivity of nervous tissue to the development of tissue stress are the following typical patterns:Neurons show high sensitivity to excitotoxicity factors, which may include some neurotransmitters, ROS, and some cytokines, especially IL-1β [241,242,243].The brain does not use higher fatty acids for energy generation, which reduces the effects of lipotoxicity factors on it. However, the brain depends critically for its energy production on aerobic glycolysis (the brain consumes ~20% oxygen under normal conditions having a mass of ~2%). Therefore, neurons are highly sensitive to glucose transport, hypoxia, and mitochondrial stress [244,245], and cognitive disorders are characteristic companions of vascular pathologies [246].The brain is isolated by the blood–brain barrier from immune and many other potentially damaging blood factors [247]. In neurodegenerative diseases, the integrity of this barrier can be compromised [248].Microglia cells are normally low-active pro-inflammatory stromal macrophages. However, their activation may play an ambiguous pathogenetic role in neurodegeneration [249,250].Most neurons are postmitotic cells, for which the typical outcomes of cellular stress are ageing, apoptosis, or programmed necrosis [251]. These processes depend not only on age [252], but also on genetic and environmental risk factors for neurodegeneration [253,254]. Therefore, neurodegenerations, for example, in normal ageing, Alzheimer’s or Parkinson’s disease display the specific characteristics of proteinopathies and their localizations [255].Sclerosis (astrogliosis) is a common feature of the different variants of neurodegeneration [256,257]. At the same time, astrocytes, as well as neurons, are subject to accelerated ageing (‘astrosenescence’), despite their relative resistance to alteration [258,259].

Pathological activation of microglial cells, in which, among other things, the TLR4/NF-κB signaling pathway is activated with the formation of NLRP3 inflammasomes, can be one of the factors of neuronal damage [260,261,262]. Inflammasome formation promotes pyroptosis, IL-1β production, and differentiation of these cells towards the M1 pole [263]. In addition, disturbances in autophagy processes in neurons and glial cells play an important role in the dysfunction of cellular stress during neurodegeneration, particularly in Alzheimer’s disease [58].

As in other para-inflammatory processes, SRs play a significant role in neurodegeneration. Thus, SR-A1 (CD204), SR-L1 (CD91, LRP1), and SR-F3 (MEGF10) are involved in the clearance of soluble amyloid proteins without evident microglia activation [264,265,266]. SR-F3 also prevent the development of secondary necrosis through participation in the uptake of apoptotic neurons by neuroglial cells [267]. Brain hypoxia can decrease the expression of SR-B1 (SCARB1) and SR-A6 (MARCO) on astrocytes, which slows down the clearance of soluble β-amyloid and increases extracellular amyloid deposition [266,268]. Conversely, the involvement of SR-B2 (CD36) and SR-J1 (RAGE) leads to pathological activation of microglia, and the involvement of SR-J1 also activates neurons [264,265]. Furthermore, in neurodegeneration, modified LDL can penetrate through the blood–brain barrier and act on neurons via SR-E1 (LOX-1), which activates the p53 transcription factor signaling pathways that promote neuronal survival or apoptosis, depending on the situation [269].

We suggest that it is reasonable to separate ageing-related neuro-parainflammation from classic neuroinflammation, such as productive inflammation, at the exacerbation stage of multiple sclerosis (MS) [270,271]. Although not all authors categorize MS as an autoimmune disease [272], the autoimmunity mechanisms have now been shown to be involved as alteration factors and causes of T-lymphocytic infiltration of demyelinating plaques [270].

### 3.6. Atherosclerosis

Currently, atherosclerosis is considered a chronic disease that can lead to various serious complications such as myocardial infarction, stroke, and other cardiovascular diseases. Inflammation and changes in lipid metabolism play a crucial role in atherogenesis, but the details of the relationship and causality of these fundamental processes remain incompletely understood [273].

From another point of view, atherosclerosis can be considered as an independent type of general pathological process, closely related to pro-inflammatory mechanisms but not identical to classical inflammation [4]. From this perspective, atherosclerosis is not a specific disease, but a common pathogenetic platform for many nosologies. The similarity between productive inflammation and atherosclerosis lies in the presence of a local macrophage accumulation formed, among other things, from monocytes migrating through the endothelial lining into the artery intima.

Cellular stress-associated classical PRRs (primarily TLR), many of the TFs (primarily NF-kB), and non-coding RNAs are involved in the differentiation of atherogenic macrophages towards M1 and their transformation into foam cells [274,275]. In particular, TLR4 activation in macrophages in atherosclerosis can be linked to DAMP (e.g., the stress protein S100), and TLR4 can form functional membrane clusters with SRs: SR-J1 (RAGE) and SR-B2 (CD36) [276]. However, it is necessary to take into consideration that not only do macrophages promote the formation of complex and unstable plaques, thus maintaining the pro-inflammatory microenvironment, but their separate types (close to the M2 pole) also exhibit anti-inflammatory activity and contribute to tissue repair and remodeling and plaque stabilization [277,278,279].

At different stages of atherosclerosis, the process of atherogenesis may involve CD8+ and CD4+ T cells as well as NK cells [280]. At the same time, various subpopulations of CD4+ and CD8+ T cells usually make up the majority of human atherosclerotic plaque leukocytes [281]. The involvement of T cells (including Th1) may also be associated with an autoimmune response to modified LDL (immune recognition of peptides from apolipoprotein B) [282]. However, unlike atherosclerosis, classical autoimmune vasculitides (macrophage HLA and T-cell dependent) are associated with an inflammatory vasa vasorum response (vasa vasoritis) [283]. In normal arteries, the vasa vasorum is limited to the adventitia, but in inflamed arteries, capillaries appear in the media and intima, which contributes to the spread of classical inflammation to these tissues as well [283]. At the same time, vascular inflammatory changes (involvement of the vasa vasorum in viral infections or autoimmune processes) may progress from the adventitial side to the intimal side of the vessel, eventually complicating the associated atherosclerotic changes in the intima [284]. In general, atherosclerosis cannot be unequivocally classified as an autoimmune disease, since possible autoimmune mechanisms in atherosclerosis are very unlikely to be the dominant mechanism of tissue alteration, being just one of the components of a more complex process. In addition, atherosclerosis, as already noted, differs from classical arterial inflammation by the absence of an inflammatory response of the vasa vasorum.

On the other hand, the connection of atherosclerosis with the development of local and systemic para-inflammation processes is currently beyond doubt, more specifically:There is an obvious association of arterial atherosclerosis with aging and low-intensity systemic metabolic alteration factors [285,286,287,288].Atherosclerosis has been shown to have a strong association with endotheliosis of large arteries, including pathological activation of endotheliocytes and endothelial glycocalyx damage [289,290,291].An important role in the pathogenesis of atherosclerosis belongs to scavenger receptors, in particular: SR-E1 (LOX-1) and SR-B2 (CD36) are involved in endotheliocyte activation, whereas SR-A1 (CD204) and SR-B2 are involved in the uptake of modified LDL by atherogenic macrophages [292,293,294,295]. In addition, endotheliocytes and macrophages are atherogenically activated by SR-J1 receptors (RAGE), which recognize advanced glycationend-products (AGEs) [26,296]. In contrast, some macrophage SRs (SR-B1 (SCARB1), SR-L1 (CD91, LRP1), SR-I1 (CD163)) and vascular myocytes SR-L display antiatherogenic activity [26,297,298].Currently, it is evident that there is a relationship between atherosclerosis and morbid obesity, metabolic syndrome, and type 2 diabetes mellitus, which, in turn, are associated with chronic systemic low-grade inflammation [299,300,301,302].As for endothelial dysfunction associated with atherosclerosis, cardiovascular diseases, tissue ischemia, and hypoxia, it increases the pro-inflammatory status of various organs and the organism as a whole [123].

We thus believe that the most fundamental characteristics of atherosclerosis could be appropriately considered in the model of a special type of general pathological process. This process has both similarities with—and differences from—productive inflammation and local para-inflammation, and is pathogenetically related to systemic age-related and metabolic changes, as well as to systemic tissue stress. It would also be more correct to consider atherosclerosis and related cardiovascular diseases in the context of interactions with other inflammatory and para-inflammatory processes at the organism level.

### 3.7. Systemic Inflammation as a General Pathological Process

We believe that currently there is every reason to distinguish systemic inflammation (SI) as an independent type of general pathological process [4,21,303,304], which can be defined as follows: “Systemic inflammation is a general multi-syndromic, phase-specific pathological process evolving in systemic injury and characterized by total inflammatory reactivity of endotheliocytes, plasma and blood cell factors, connective tissue, and, at the terminal stage, microcirculatory disorders in vital organs and tissues” [4]. The key pathogenetic feature of SI is an organism-wide microvascular inflammatory response comparable in severity to that in the focus of classical inflammation. The culmination of acute SI has characteristic clinical signs in the form of refractory shock, coagulopathy of the disseminated intravascular coagulation type, and rapidly progressive multiple organ failure [305,306]. However, there is a major problem of the initial, marginal manifestations of SI, which must be diagnosed in time and differentiated from the systemic signs of other pro-inflammatory processes.

The greatest difficulty is the differentiation between systemic inflammation and systemic inflammatory response, which consists in the accumulation of pro-inflammatory mediators in plasma [307]. It should be noted that cytokinemia and other manifestations of SIR can be unequivocally indicative of SI only in some cases, taking into account the phases and other features of the process. Note further that SIR can also be quite intense in some (hyperergic) variants of classical inflammation. Thus, integral criteria based on at least 3–5 specific SIR indicators, including pro-inflammatory and anti-inflammatory cytokines, are required to identify specific clinically and pathogenetically relevant levels of SIR [307]. A more precise verification of individual SI phases would require even more complex criteria that should include—according to a certain algorithm and in addition to SIR level determination—criteria for systemic alteration, coagulopathy, organ dysfunction, neuroendocrine distress reaction, microcirculatory changes (for example, according to vital tissue microscopy), and other characteristic signs of SI [308,309].

An even more difficult problem is posed by the verification and determination of the clinical and pathogenetic significance of chronic SI, which has no clear clinical equivalents. Meanwhile, as we have already noted [123], the systemic changes in the pro-inflammatory status of some patients with autoimmune pathology, end-stage renal disease, and some other severe chronic diseases go far beyond the available understandings of the pathogenesis of classical inflammation and systemic low-grade inflammation.

We believe that comprehensive characterization of SI and its differentiation from the systemic manifestations of other general pathological processes is one of the most important challenges in modern practical and fundamental medicine. For the moment, we will limit ourselves to a brief remark in order to discuss the current state of SI research in more detail later, in separate publications of this Special Issue.

## 4. Evolutionary Trends in the Development of Inflammation

Understanding of the evolutionary patterns in the emergence and development of mechanisms of inflammation and innate and adaptive immunity is important for the holistic characterization of inflammation as a general pathological process. The following stages may be distinguished in the development of the inflammatory process in the evolution of species:The development of tissue pro-inflammatory stress based on non-adaptive, innate immunity mechanisms is characteristic of all metazoans. Thus, invertebrates have all the basic protective mechanisms of phagocytes, including: a variety of PRRs, hydrolases, free radicals, cationic proteins, extracellular DNA traps, etc. [310]. For example, compared to mammals, some echinoderm species have about an order of magnitude greater variety of extracellular and intracellular PRRs of the two most important families, TLR and NLR [311]. Invertebrates, as well as vertebrates, also have the problem of immune system ageing related, among other things, to cellular stress mechanisms [312]. Another variant of the cell/tissue stress outcome—i.e., tumorigenesis—is also characteristic of invertebrates [313].Highly organized invertebrates that have hemocytes, hemolymph, and a neuroendocrine system are capable of responding to damage and infection by developing SIR, which consists in the accumulation of stress hormones and neurotransmitters [314], some hemocyte populations and a variety of bactericidal and pro-inflammatory molecules, including cytokine-like factors in hemolymph [315,316,317,318,319,320,321]. The hemostasis system in invertebrates is not specialized and is mainly represented by cells (hemocytes) and adhesive molecules of the immune system [322,323]. In some invertebrates, such as insects, the innate immune system is capable of adaptive responses that usually provide a short-term acquired resistance to viral and extracellular infections [324,325,326].The development of classical inflammation and the emergence of the lymphocytic adaptive immune system and the progressive hemostasis system became possible only in vertebrates due to the emergence of an elementary basis of microcirculation in them—microcirculatory units including vascular (precapillary arterioles, capillaries, capillary sphincters, postcapillary venules) and extravascular transport communications that ensure exchange processes between blood and a particular tissue area [327,328]. This determined not only the possibility of directed and selective leukocyte migration, but also the appearance of interrelated components of the exudative vascular complex (EVC), including the microvascular network, mast cells, and complement, kininogenesis, and hemostasis plasma systems (Table 3). Vertebrates starting with bony fish reveal orthologues of major TFs and cytokines that are specific to different T-cell immune response vectors (i) [329,330] and provide the development of specific productive inflammation directed towards a particular infectious factor [329,330].

4.The EVC and immune system of a more advanced level in higher vertebrates (reptiles, birds, and mammals) are responsible for the possibility of development of the most traumatic variants of exudative–destructive inflammation such as caseous necrosis and, in mammals, purulent inflammation (Table 3). At present, it can be confidently stated that systemic inflammation (systemic ‘inflammatory microcirculation’) can occur only in mammals and isolated manifestations of systemic inflammation may be found in birds.

These evolutionary differences seem to manifest themselves most clearly in the development of sepsis. Thus, systemic bacterial infections in fish and amphibians are characterized by microbial colonization of the gill, skin, muscles, internal organs, as well as their erosions, ulcerations, necroses, vascular damage, and hemorrhages [339,340,341]. Amphibians additionally show a more intense exudative reaction with fibrin accumulation in infected organs [342]. In reptiles, the generalization of infection is associated with multiple granulomas in internal organs, and with a large number of heterophils (mammalian neutrophil analogues) in granulomas during extracellular bacterial infections and predominantly lethal lesions of the heart and central nervous system [343,344]. In birds, generalized infection also presents as secondary microbial colonization; often affects the endocardium and myocardium; and reveals fibrinous deposits in tissues and granulocytic infiltrates, the major causes of death being thromboembolism in vital organs and septic endocarditis [343,345,346]. In humans, however, secondary microbial colonization of internal organs, and even bacteremia, is not a necessary condition for death in sepsis [347,348]. In fact, the association between bacteremia and endotoxemia is not always detected in patients [349]. In dogs, cats and various other mammalian species, the typical features of infectious and aseptic critical states are the accumulation of cytokines and other phlogogenic factors in the blood, systemic microthrombosis and microvascular activation during shock and multiple organ dysfunction, in some cases without signs of secondary pyemia [350].

Thus, the processes of para-inflammation, different variants of classical inflammation, and life-critical systemic inflammation have arisen at different stages of evolution. This evolutionary division is an additional argument for the need to comprehensively describe the general patterns of these processes, but also to differentiate these typical pathological processes in humans.

## 5. Discussion

When conducting biomedical research, there is a need to characterize not only study objects and subjects (patients), specific clinical definitions, and experimental models, but also the processes under study as integral phenomena. Principal models of typical or general pathological processes should be responsible for the common characterization of the main pathological processes. However, at the moment, the problem of general pathological processes itself has left the scientific discourse and scientific publications, remaining the domain of textbooks on pathological physiology and general pathology in an immobilized, irrelevant form. At the same time, conceptual syndromes are often used as surrogates of general pathological processes, as noted above, but they are neither full-fledged protocol clinical definitions, nor full-fledged models of typical pathological processes. This determines, in our opinion, the clear predominance of analytical research over synthetic ones, and this will inevitably lead to the accumulation of internal contradictions in the system of knowledge of both fundamental and practical medicine.

Given the above, our work was aimed at substantiating the necessity of modernizing and updating the theory of general pathological processes. We believe that general pathological processes should not be considered separately, but in a unified system in which the key system-forming factors are cellular stress (elementary functional unit of various pathological processes) and tissue pro-inflammatory stress. On this basis, many pathological and some physiological processes have common molecular mechanisms that manifest themselves in different contexts. This allows processes different in nature and pathological manifestations to be integrated into more holistic systems. This conceptual approach is presented in the most fundamental form in Figure 8, and in a more detailed form in Figure 9.

It is worth noting that the theory of general pathological processes is the foundation for the construction of clinical definition models. Practical medicine, in turn, reflects the likelihood degree of theoretical models of general pathological processes.

It should be borne in mind, however, that implementing this approach will require substantial changes in the system of scientific knowledge in various fields of medicine, including:The use of clinical criteria alone is insufficient for the verification of complex pathological processes, e.g., metabolic syndrome criteria for the verification of ChSLGI, or Sepsis-3 and SIR criteria for the verification of systemic inflammation as a general pathological process.In molecular research in an in vitro system, there will be a need for a more fundamental characterization of the cellular and tissue system of which the molecular mechanisms under study are a part.The development and practical use of clinical models will need to be harmonized with models of general pathological processes, which will objectively determine stricter requirements for theoretical training not only for scientific researchers, but also for practitioners.New scientific disciplines will probably need to be created, or existing disciplines—such as systems biology and integrative medicine—may need to be substantially modernized.The key objective of modern medicine, according to many specialists, should be the prevention of diseases and their complications, which will require a theoretical substantiation of the relationship between physiological and pathological processes, and characterization of transition zones between qualitatively different human pathological states.Pathology assessment methods will require additional sophistication along with broader use of computer network information technologies, including clinical decision support software.

A question then arises: how appropriate is this modernization for addressing specific problems in practical health care? It should be noted that this kind of change may come around as a result of being unable to live the old way rather than breaking new grounds. From this perspective, we can identify several crisis trends in current practical and theoretical medicine which, in our opinion, may over time become prerequisites for the advancement of this kind of systematization of knowledge, in particular:The distance between the avalanche-like accumulation of primary research data and their synthesis between analytical and synthetic approaches in medicine is increasing.In etiopathogenetic therapy, the famous Hippocratic postulate “Primum non nocere” (“First, do no harm”) is often violated. Applying the principles of evidence-based medicine limits these negative effects. However, this problem, too, will inevitably require a revision of many theoretical concepts, primarily through wider use of the systems approach in medicine.At first glance, a systematic approach based on the use of models of general patterns of pathology contradicts the principles of a personalized approach in medicine, which is not true. On the contrary, it is impossible to describe a specific clinical situation and propose a patient-centered treatment protocol without separating the general from the particular. Thus, the use of a personalized approach will, over time, increasingly often ‘stumble’ over the unresolved general problems of pathology.The ever-increasing specialization of clinicians makes it difficult for them to assess the patient’s body as a holistic system. Often, there is a lack of cooperation among the various practitioners providing care for a particular patient at the same time.The prognostic, diagnostic, and outcome monitoring criteria used in clinical protocols already lag behind the capabilities of modern technology, including data from molecular research, and instrumental and information technologies. The idea that this problem can only be solved by using mathematical methods, ignoring heuristic approaches to the modeling of complex systems is, in our opinion, erroneous for many reasons. The widespread use of the terms: ‘inflammation’, ‘neuroinflammation’, ‘systemic inflammation’, ‘systemic inflammatory response’, etc. without their necessary characterization has essentially turned these terms into ‘vague’ concepts that require specification in each study.

## 6. Conclusions

The present review is one possible step towards solving the problem of unclear characterization of general pathological processes, the models of which provide a basis for understanding various human pathologies. A common pathological platform for general pathological processes is pro-inflammatory tissue stress as a tissue response to various injuries. Cellular stress, in turn, is an elementary but holistic functional unit of tissue stress and, consequently, of general pathological processes. Moreover, since tissue stress may also develop under a number of physiological conditions, our approach allows us to describe the transition states between norm and pathology, as well as the transformation of one pathological process into another, which is important for solving problems in clinical practice. At the same time, various forms of inflammation are the most typical but not exclusive variants of tissue stress.

## Figures and Tables

**Figure 1 ijms-23-04596-f001:**
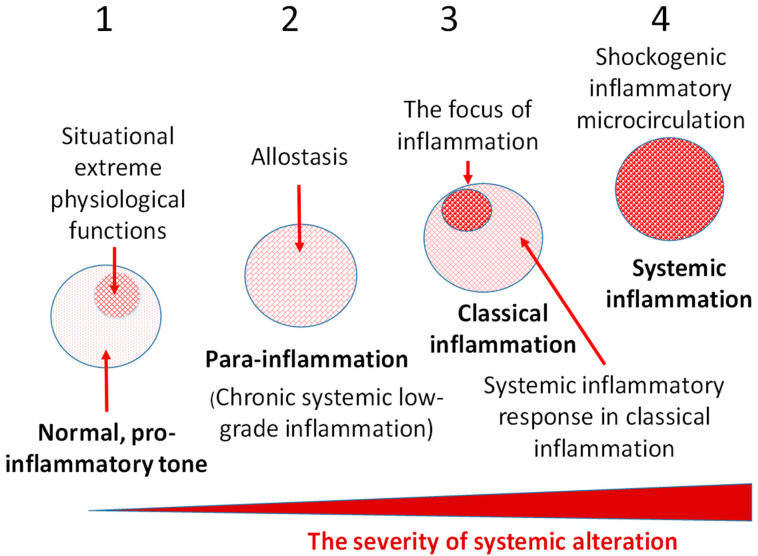
Variants of tissue pro-inflammatory stress. 1—Physiological variants of TS; 2—Non-classical low-grade inflammation (para-inflammation), which at systemic level may be manifest as stably altered homeostasis (allostasis); 3—Classical inflammation (the organism’s response to a significant local injury) is characterized by the presence of its attribute—a focus of inflammation and, in some cases, a systemic inflammatory response aimed at resourcing the focus of inflammation; 4—Life-critical systemic inflammation, the key phenomenon of which is a systemic microvascular response comparable in intensity to the local response in the focus of classical inflammation.

**Figure 2 ijms-23-04596-f002:**
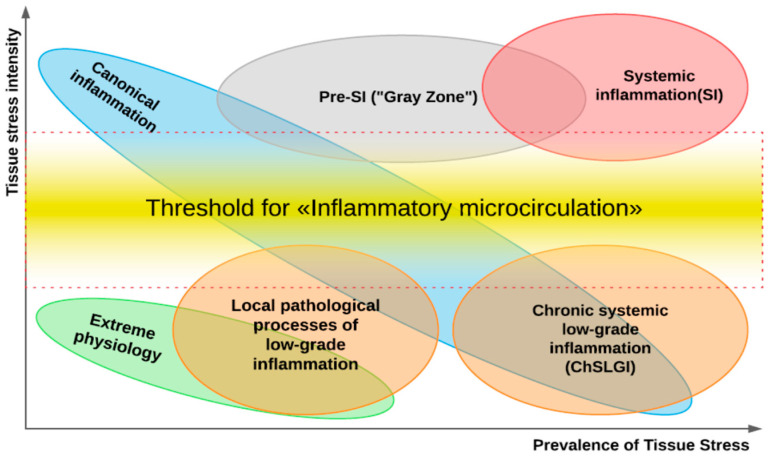
Tissue stress and general pathological processes (from Gusev E. et al., 2021). Note: The ratio of intensity to prevalence of damaging factors initiating a ‘response’ in the form of tissue pro-inflammatory stress—a common pathogenetic underpinning of all pathological processes—can be used to distinguish three ‘big’ general pathological processes (classical inflammation, systemic inflammation, and ChSLGI). The figure shows that the systemic manifestations of classical inflammation and ChSLGI may be comparable in terms of the localization and intensity of pro-inflammatory responses, requiring additional diagnostic methods to separate them.

**Figure 3 ijms-23-04596-f003:**
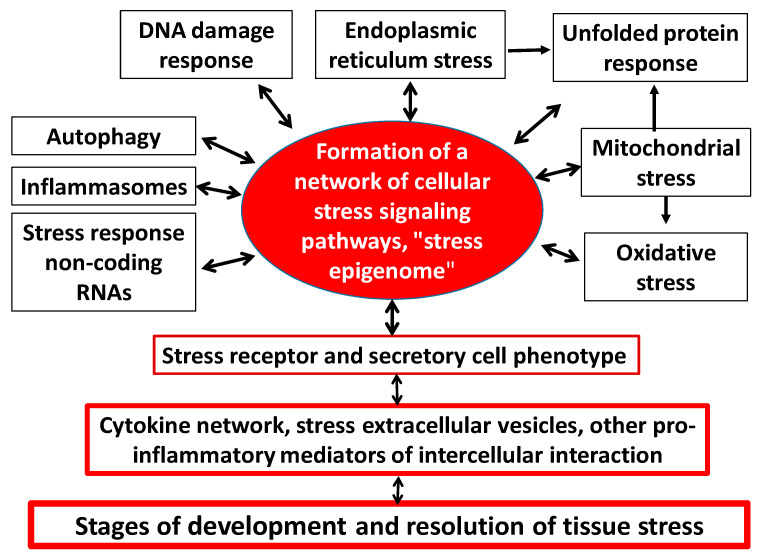
Structure of typical cellular stress processes and its relationship with tissue stress.

**Figure 4 ijms-23-04596-f004:**
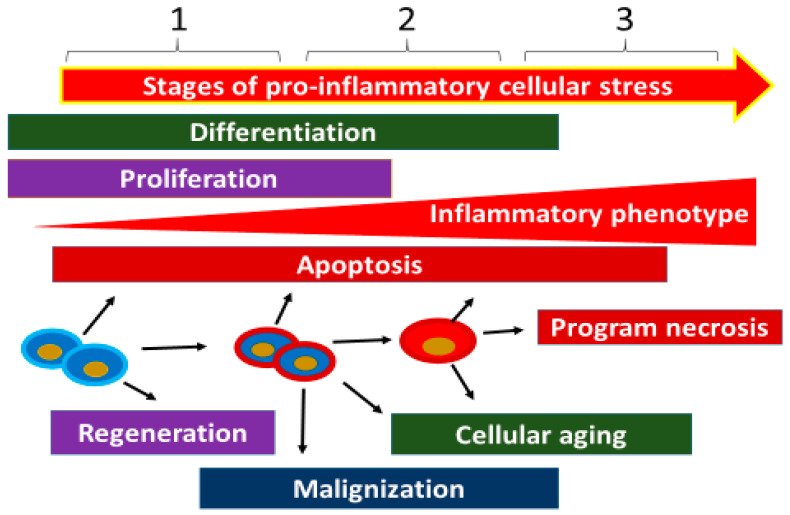
Three stages of cellular stress development. Stage 1 is typical for proliferating cells; it is characterized by the predominance of growth factors in the secretory phenotype; relatively moderate manifestations of pro-inflammatory phenotype (including oxidative stress); dominance of anabolic processes; and adaptation to the moderate action of damaging factors. This stage can be complicated by the processes of tissue metaplasia and malignization. At the level of tissue stress, this stage is also typical for many physiological and pathology borderline processes, as well as for the repair (regenerative) stage of inflammation. Stage 2 is a transitional stage; it is characterized by different proportions between the first and third stages. Stage 3 is characterized by more pronounced manifestations of the pro-inflammatory phenotype in response to the increasing effect of damaging factors; increasing insulin resistance; cell cycle blockade; accelerated cell aging; an increasing role of autophagy and mitochondrial stress; and a high probability of programmed necrosis in the variant of pyroptosis, NETosis, and necrobiosis. When microvessels and migrating leukocytes are involved in these processes, conditions emerge for the formation of a canonical inflammation focus or for the development of systemic microcirculatory disorders as signs of systemic inflammation.

**Figure 5 ijms-23-04596-f005:**
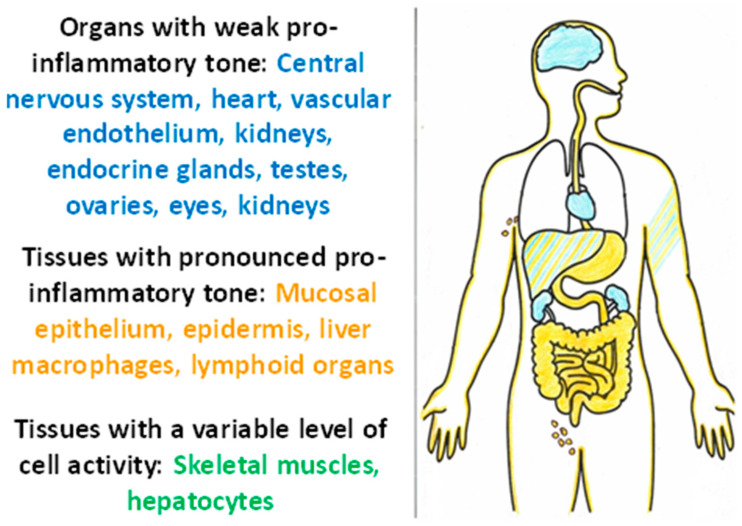
Organs with varying degrees of tissue stress under physiological conditions.

**Figure 6 ijms-23-04596-f006:**
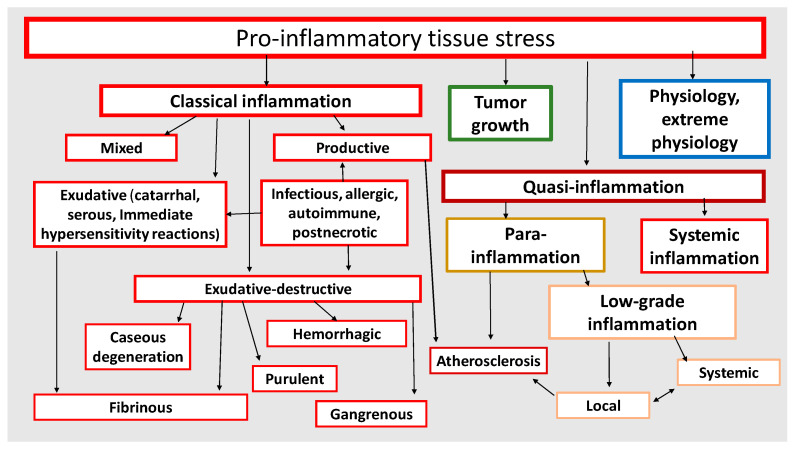
Pro-inflammatory tissue stress as a common basis for the development of general pathological and some physiological processes.

**Figure 7 ijms-23-04596-f007:**
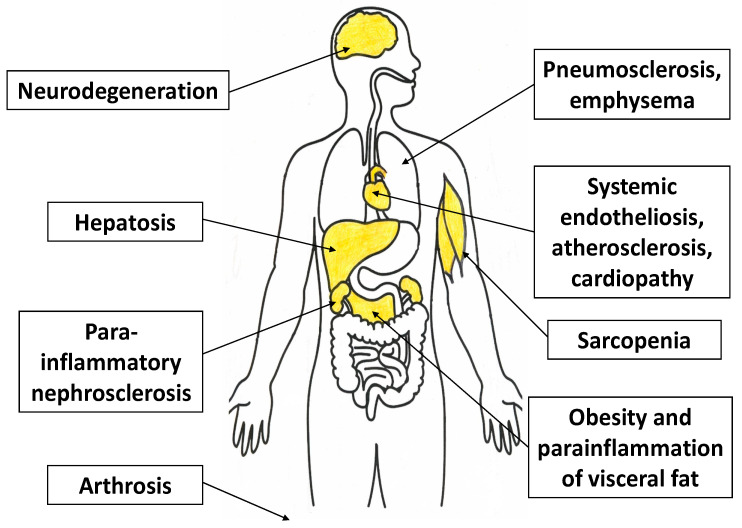
The main target organs in the development of chronic systemic low-grade inflammation.

**Figure 8 ijms-23-04596-f008:**
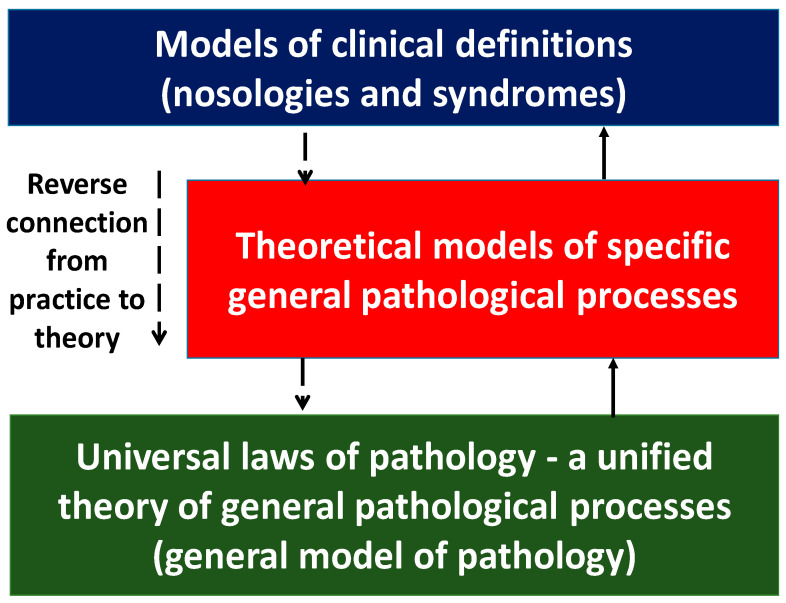
Conceptual relationships between theoretical and clinical definitions.

**Figure 9 ijms-23-04596-f009:**
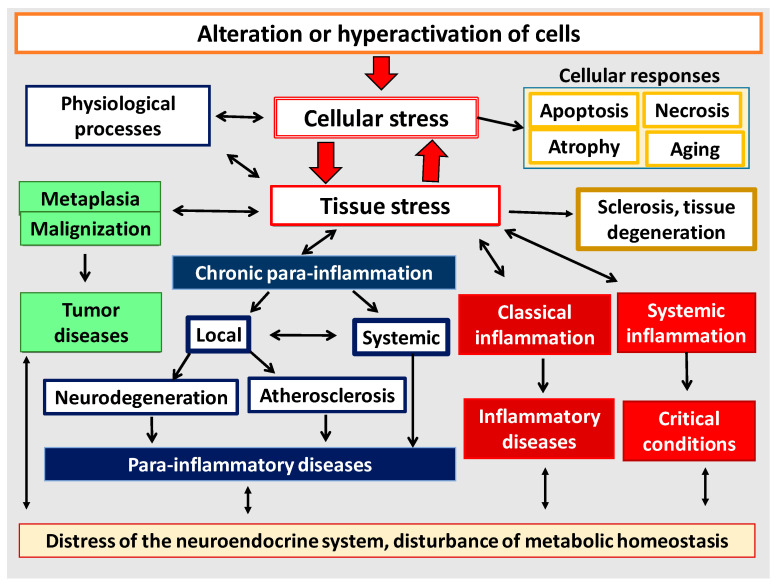
A principal system for the relationships between tissue pro-inflammatory stress and key general pathological processes.

**Table 1 ijms-23-04596-t001:** Some phenomena of cellular stress characterizing possible stages of its development.

Phenomena	Stage 1	Stage 2	Stage 3
Proliferation	activated	variable	suppressed
Dominance ^1^ of growth factors	yes	no	no
Insulin resistance	no	possible	yes
Phosphoinositide 3-kinases (PI3K) ^2^	activation	not typical	not typical
mTORC1 expression	high	variable	variable or low
Autophagy	low	elevated	high
Inflammasomes	low	NLRP3 activation in various cells	
Apoptosis	possible	possible	possible
Programmed necrosis ^3^	not typical	unlikely	possible
Effects of SR on PRR	suppressed	variable	activated
Purinergic receptors ^4^	P1	P2X and P2Y	P2X and P2Y
p53/NF-κB ratio	↑/↓	↓/↑	↓/↑
Mitogen activated protein kinases	ERK > JNK and p38	ERK < JNK and p38	ERK < JNK and p38
Production and reception of pro-inflammatory cytokines	moderate	high	Unstable ^4^
iNOS endotheliocytes	inactive	inactive	active
cNOS endotheliocytes	?	inhibited	?
Unfolded protein response	progression		
ROS formation	progression		
NF-κB, AP-1, HIF-1α, HSFs, Egr	progression of expression of these transcription factors		
The role of non-coding RNA	depends on cell type and formation of extracellular vesicles		

Note: It is the author’s integral table compiled as a result of the analysis of numerous literature data presented in the text of Section 2.5. ^1^—in the cytokine spectrum; ^2^—PI3K, which is dependent on insulin and many growth factors; ^3^—pyroptosis, necroptosis, NETosis, autophagic cell death; ^4^—main ligands: for P1—adenosine, for P2—ATP; ↑/↓—more/less; SR—scavenger receptor; PRR—pattern recognition receptor; ROS—reactive oxygen species; NOS—NO synthase: i—inducible and c—constitutive.

**Table 2 ijms-23-04596-t002:** Vectors of immune response (I) [163,164,165,166,167,168,169,170].

I	Th (TFs), Cytokines: Activators and *-Inhibitors	Main Cytokines Th	Other Cells (TFs; Cytokines Production // Reception; *-Inhibitors)	Major Role in Inflammation	Complications
**i1**	**Th1** (T-bet, STAT4, STAT1); IL-12, IFN-γ; IL-4 *, IL-10 *	IFN-γ, IL-2, CXCL10, CXCL11	M1 (STAT1, NF-κB; TNF-α, IL-1β, IL-6, IL-12, IL-15, IL-23 // IFN-γ, TNF-α; IL-10 *, TGF-β *), CTL, NK, ILC1 (IFN-γ)	Response to intracellular infection, antitumor immunity	Autoimmune processes, allograft rejection
**i2**	**Th2** ^1^ (GATA3, STAT5, STAT6); IL-4, IL-25, IL-33; IFN-γ *, TGF-β *, IL-12 *	IL-4, IL-5, IL-13, IL-25, CCL17, CCL22	M2a (STAT6, STAT1, GATA3; IL-6, IL-10 // IL-4, IL-13, IL-33), Tc2 (IL-5, IL-13), mast cells, basophils, ILC2 (IL-4), epithelial cells, eosinophils	Antimetazoan immunity, chronic inflammation, inflammation in damage-sensitive tissues	Allergic processes, i1 suppression, tissue fibrosis
**i3**	**Th17** (RORγt, RORα, STAT3, STAT5); IL-1β, IL-6, IL-23, TGFβ; IL-10*	IL-17A/F, IL-21, IL-22, CCL20, CXCL-1,7,20	M2b (TNF-α, IL-1β, IL-6, IL-10 // IL-17A/F, TNF-α, IL-1, IL-6, IL-23; IL-10 *), Tc17 (IL-17), neutrophils, ILC3	Response to extracellular infection	Autoimmune processes, allograft rejection
	**Th22** (RUNX3, AHR, STAT3); IL-6, IL-1β, TNF-α	IL-22, CCL-2, 20, CXCL-9, 10, 11, FGF	Epithelial cells, langerhans cells	Protection of the epidermis against extracellular infection	Autoimmune skin processes
**i-reg**	**Treg** (FOXP3, STAT3/5, SMAD2/3, RORγt GATA3,); IL-2, IL-10, TGF-β	TGFβ, IL-10, CCL4	M2c (SMAD2, SMAD3, STAT3; IL-10, TGFβ // IL-10, TGF-β), Tr1 (IL-10, IFN-γ), Tc-reg (TGFβ, IL-10), ILC10 (IL-10)	Limiting the expression of i1 and i3, inhibition of the autoimmune process	i1 and i3 immunosup- pression, infections, tumor growth

Note: *—inhibitors of immune response; TFs—transcription factors (the main TFs are underlined); Th—CD4+ T-helper; CTL—cytotoxic T lymphocytes, or Tc1; NK—natural killer cells; Tc—CD8+ T cells; Treg—CD4+ regulatory T cells; ILC—innate lymphoid cells; Tr1—Type 1 regulatory T cells (CD4+); ^1^ some authors categorize into i2 also Th9, which are induced by TGF-β and IL-4 from Th2 precursors (the main TF is PU.1), are major producers of IL-9, contribute to anti-tumor immunity (in contrast to Th2), but may also participate in autoimmune processes [164,171].

**Table 3 ijms-23-04596-t003:** Evolutionary patterns of inflammation and immunity [331,332].

Immune and Inflammatory Mechanisms	Taxa				
	Invertebrates	Bony Fishes	Reptiles	Birds	Mammals
The reaction of phagocytes	Yes ^1^	yes	yes	yes	yes
PRR in phagocytes	yes	yes	yes	yes	yes
Lymph formation ^2^	no	yes	yes	yes	yes
The lymph nodes	no	no	no	yes/no	yes
Vessels, hearts	yes/no	yes	yes	yes	yes
Blood microcirculation	No ^3^	yes	yes	yes	yes
Exudative reactions	no	yes	yes	yes	yes
Histamine in mast cells	no	yes/no ^4^	yes	yes	yes
Anaphylatoxins(C3a, C5a)	no	yes	yes	yes	Yes ^5^
Kinins	no	yes	yes	yes	yes
Kallikrein–kinins	no	no	yes	yes	yes
Hemostasis system	no	yes	yes	yes	Yes ^6^
Non-nucleated platelets	no	no	no	no	yes
Adaptive immunity	yes/no	yes	yes	yes	yes
Lymphoid system	no	yes	yes	yes	yes
Cytokine network	No ^7^	yes	yes	yes	yes
Main classes Ig	no	IgM	IgM, IgY	IgY, IgM	IgG, IgM
IgE	no	no	no	no	yes
Delayed-type hypersensitivity	no	no	no	yes/no ^8^	yes
Autoimmune processes	no	yes	yes	yes	yes
Para-inflammation	yes	yes	yes	yes	yes
Classical inflammation	no	yes	yes	yes	yes
Purulent inflammation	no	no	no	no	yes
SIR	yes/no	yes	yes	yes	yes
Systemic inflammation ^9^	no	no	no	?	yes
NES distress reaction ^10^	?	?	?	yes	yes

Note: yes—presence of a sign; no—absence of a sign; yes/no—sign detected in individual species; “?”—no reliable data on the phenomenon as a whole, but individual manifestations are possible; PRR—pattern recognition receptors; Ig—immunoglobulin; SIR—systemic inflammatory response; NES—neuroendocrine system; ^1^—e.g., parasite encapsulation [310]; ^2^—separation of lymph and blood; ^3^—the absence of a system of microcirculatory units; ^4^—in the most evolutionarily developed fish [333]; ^5^—only in mammals, complement anaphylatoxins (C3a and C5a) are formed in the liquid phase of the blood, for example, under the influence of hemostasis factors (XIIa, plasmin and thrombin) [334]; ^6^—only mammals have an extrinsic pathway for hemostasis activation (associated with the appearance of binding factor XI in them), and there are significantly fewer triggering factors (V, VII, and a soluble form of tissue factor) in plasma in birds than in mammals [335]; ^7^—in some invertebrates, some cytokine-like factors may be detected in hemolymph and other tissues, but there is no developed cytokine network; ^8^—DTH in birds is associated with the presence of high-affinity Fc receptors to IgY (FcυR) on mast cells [336], but DTH is significantly slower in birds than in mammals; ^9^—in this case, systemic inflammation is seen as a general pathological process with a systemic ‘inflammatory microcirculation’ phenomenon, not as a synonym for SIR; ^10^—according to the theory of G. Selye [337,338].

## Data Availability

Not applicable.

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
