# Peer review of "Inflammation: A New Look at an Old Problem"

_ijms, 2022, doi:10.3390/ijms23094596_

Round 1

Reviewer 1 Report

Inflammation: A New Look at an Old Problem, IJMS_1673663

The manuscript from Gusev et al. reviews different type of inflammation

The authors focus on a scientifically important field. Structure of the paper is logic and well organized. Due to the high mortality rate of chronic inflammation and the need to further study that field manuscript fits in the scope of IJMS journal.

I have the following major comments:

  1. The paper seems to be more philosophic and conceptual rather than review of the latest results about different types of inflammation. The manuscript style is closer to textbook stylistic and too long covering to many aspects for a focused review. That is the subjective criticism of the Reviewer and not a claim to change the entirely text. However, I highly recommend the implementation of the latest research articles on the filed of inflammation, the latest experimental data should be included at least 10-12 research articles from 2021-2022 about unresolved inflammation or focused up to the authors’ choice.
  2. Autoimmunity is at least frequent and devastating as cancer or other chronic inflammatory conditions of devastating disease. Please, include a subchapter about inflammatory conditions caused by autoimmune reactions.

Minor comments:

  1. Line 490: ‘may be’ is repeated in the sentence
  2. Abbreviations should be explained at first appearance (do not need to explain again e.g. ChSLGI in line 118. I do not list all cases but please check the manuscript for these situations for all abbreviations

Author Response

Dear Reviewer,

Thank you very much for your kind response with your valuable advice. We enclose our corrected review manuscript with all highlighted changes.

Our explanations of the corrections in response to your comments:

  1. “The paper seems to be more philosophic and conceptual rather than review of the latest results about different types of inflammation. The manuscript style is closer to textbook stylistic and too long covering to many aspects for a focused review. That is the subjective criticism of the Reviewer and not a claim to change the entirely text. However, I highly recommend the implementation of the latest research articles on the filed of inflammation, the latest experimental data should be included at least 10-12 research articles from 2021-2022 about unresolved inflammation or focused up to the authors’ choice”.

We agree with the reviewer's opinion. However, this peculiarity of the review is caused by the following reasons. We believe that the unresolved general pathology and pathological physiology problems are objective obstacles to the solution of many medical practice problems. The theory of general pathological processes in general and inflammation in particular, presented in modern textbooks, is outdated and needs significant modernization. A consequence of this is the appearance of numerous models of conceptual syndromes, which are surrogates of general pathological processes, but are not able to fulfill their function. Our paper substantiates the necessity of creating a new theory of general pathological processes, which should become a theoretical basis for the construction of more private models of clinical definitions. Because of the broad scope of the modern medical problems, the paper became conceptual and partly philosophical.

The original version of the review had 56 references for 2021 and 10 references for 2022. At the same time, considering the reviewer's opinion, we have added 24 more references for these years.

  1. “Autoimmunity is at least frequent and devastating as cancer or other chronic inflammatory conditions of devastating disease. Please, include a subchapter about inflammatory conditions caused by autoimmune reactions”.

We have added an additional subsection (3.2.) devoted to autoinflammatory and autoimmune variants of inflammation. At the same time, based on the goals and large total volume of our review, we were able to focus only on the most general characteristic of the autoimmune process.

  1. “Line 490: ‘may be’ is repeated in the sentence”.

Repeated phrase is deleted.

  1. “Abbreviations should be explained at first appearance (do not need to explain again e.g. ChSLGI in line 118. I do not list all cases but please check the manuscript for these situations for all abbreviations”.

Redundant explanations of abbreviations have been removed; at the end of the paper we have added a list of abbreviations for the reader's convenience.

Reviewer 2 Report

The review by Gusev and Zhuravlev proposes a new paradigm on the interpretation of pathologic mechanisms, which are considered as deeply interrelated and possibly united in a new paradigma of diseases interpretation. The manuscript is well written and clearly presented although it is quite intense and dominated by the new interesting point of view on the study of pathology and inflammatory processes.

Author Response

Dear Reviewer,

We thank you very much for reviewing and appreciating our manuscript.

Reviewer 3 Report

Title: "Inflammation: A New Look at an Old Problem” 

Authors: Evgenii Gusev, Yulia Zhuravleva

Comments:

The objective of this literature review is to make theoretical arguments for the need for an up-to-date theory of the relationships between major pathological processes in humans based on the integrative role of molecular mechanisms of proinflammatory stress in cells and tissues.

A common thread and clear structures and connections are missing here or are not made clear. This is probably due to a lack of focus (already due to the title) and a much too broad topic.

Major points:

  1. The topic is altogether much too broad, the focus is partly completely lost
  2. - The abstract should be focused and limited to a maximum of 250 words.
  3. English and formatting are very expandable
  4. Many abbreviations, but no list of abbreviations
  5. Page 2 line 93: how are "classical" and "non-classical" forms of inflammation defined?
  6. In many chapters and in many places there are bullets and numbers in the middle of the chapter, which (in this quantity) disturbs the reading flow.
  7. A brief legend to Figure 4 would be helpful
  8. Especially within chapter 3, which is actually about "tissue stress", the focus is often lost: on the one hand, the focus often changes from "tissue" to details of "inflammation", in general, so many different subchapters and ADDITIONAL subdivisions and lists are used that one often loses track of what it is all about.
  9. Also, the first level of the subchapters of chapter 3 are each completely inconsistently worded
  10. I miss transitions and connections as well as a red thread in the review, here it seems more like an unfiltered listed flood of information.
  11. Page 17, line 634-644: I miss the mention of the important term "tumor microenvironment", which is significantly involved in inflammation propagation in tumor cells and has such a great importance.
  12. Page 17, lines 657-659: This is drawing its own conclusion that has been known for a long time. There are numerous papers that published the relationship between cancer and inflammation as an overall concept years ago.
  13. Rather, the present conclusion should be called a discussion, and it should be referred to. A separate conclusion summarizing the main message of the paper and its scientific relevance in 3-5 sentences would be helpful to the reader.
  14. References (the biggest construction site of the paper!).

- The first reference is in line 68. What about the statements made earlier?

- Lines 83-85: What is the source of the quotation?

- There are very long passages where there is no source citation in between, e.g. lines 156-167, 238-269, 279-293, 323-343, 363-376, 424-435, 481-497, 534-557, 562-574, 664-695, 857-869. This should be corrected.

  1. Page 5, Chapter 2.3: Only blanket references are given here on line 178 and not on the entire rest of the page.
  2. Only three references are given on the entire page 6. These should be better broken down and stated correctly.
  3. In Tables 1, 2, and 3, the references are not given at all, are incomplete, or are blanket references. The summarized information cannot be assigned to an origin for the reader.
  4. Page 25, line 980: Please insert the reference paper for the cited Selye theory.

Author Response

Dear Reviewer,

Thank you very much for your painstaking work, careful reading of our paper, and for your valuable comments and advices. We enclose our corrected  manuscript with all highlighted changes.

Our explanations of the corrections in response to your comments:

Point 1:“The topic is altogether much too broad, the focus is partly completely lost”.

Response 1: We are receptive to the reviewer's opinion. However, this peculiarity of the review is caused by the following reasons. We believe that the unresolved general pathology and pathological physiology problems are objective obstacles to the solution of many medical practice problems. The theory of general pathological processes, in general, and inflammation in particular, presented in modern textbooks, is outdated and needs significant modernization. A consequence of this is the appearance of numerous models of conceptual syndromes, which are surrogates of general pathological processes, but are not able to fulfill their function. Our paper substantiates the necessity of creating a new theory of general pathological processes (on the basis of the binding role of proinflammatory tissue stress), which should become a theoretical basis for the construction of more private models of clinical definitions. Thus, the focus of the paper was very broad and, in some places, also unclear for quite objective reasons (there are still many unresolved problems in this knowledge area). However, we have tried to take into account all of the reviewer's comments and thereby improve the understanding of our review content.

Point 2: “The abstract should be focused and limited to a maximum of 250 words”.

Response 2: We have reduced the abstract to the required size.

Point 3: “English and formatting are very expandable”.

Response 3: English was corrected by a professional linguist with a long experience from a specialized proofreading organization.

Perhaps the specifics of the writing are caused by the high level of conceptual generalizations.

Point 4: “Many abbreviations, but no list of abbreviations”.

Response 4: We have added a list of abbreviations at the end of the paper.

Point 5: “Page 2 line 93: how are "classical" and "non-classical" forms of inflammation defined?”

Response 5: We have made the required clarification.

Point 6: “In many chapters and in many places there are bullets and numbers in the middle of the chapter, which (in this quantity) disturbs the reading flow”.

Response 6: In our experience, when presenting this kind of information, the absence of separate subsections in the chapter can make it difficult for readers to understand this information. We tried to minimize the number of subsections, but we were forced to add another subsection (3.2) on autoimmunity at the request of one of the reviewers.

Point 7: “A brief legend to Figure 4 would be helpful”.

Response 7: We have added a note to Figure 4.

Point 8: “Especially within chapter 3, which is actually about "tissue stress", the focus is often lost: on the one hand, the focus often changes from "tissue" to details of "inflammation", in general, so many different subchapters and ADDITIONAL subdivisions and lists are used that one often loses track of what it is all about”.

Response 8: We have added some linking sentences to the text of Chapter 3, as well as differentiated the terms "inflammation" and "proinflammatory tissue stress" in more detail.

Point 9: “Also, the first level of the subchapters of chapter 3 are each completely inconsistently worded”

Response 9: We have tried to improve the writing of subchapters 3.1.

Point 10: “I miss transitions and connections as well as a red thread in the review, here it seems more like an unfiltered listed flood of information”.

Response 10: We have added linking sentences to all the chapters of the paper.

Point 11: “Page 17, line 634-644: I miss the mention of the important term "tumor microenvironment", which is significantly involved in inflammation propagation in tumor cells and has such a great importance”.

Response 11: We have expanded on the role of the tumor microenvironment in the relevant section of the paper.

Point 12: “Page 17, lines 657-659: This is drawing its own conclusion that has been known for a long time. There are numerous papers that published the relationship between cancer and inflammation as an overall concept years ago”.

Response 12: We have clarified our position on this issue and inserted some references to the relevant papers.

Point 13: “Rather, the present conclusion should be called a discussion, and it should be referred to. A separate conclusion summarizing the main message of the paper and its scientific relevance in 3-5 sentences would be helpful to the reader”.

Response 13: We have changed the Conclusion section and renamed it Discussion, taking into account the reviewer's comment. Also, we have added a brief conclusion.

Point 14: “References (the biggest construction site of the paper!).

- The first reference is in line 68. What about the statements made earlier?

- Lines 83-85: What is the source of the quotation?

- There are very long passages where there is no source citation in between, e.g. lines 156-167, 238-269, 279-293, 323-343, 363-376, 424-435, 481-497, 534-557, 562-574, 664-695, 857-869. This should be corrected”.

Response 14: We have added additional references to the text of the paper, where they were missing.

Point 15: “Page 5, Chapter 2.3: Only blanket references are given here on line 178 and not on the entire rest of the page”.

Response 15: We have added additional references in section 2.3.

Point 16: “Only three references are given on the entire page 6. These should be better broken down and stated correctly”.

Response 16: We have made the arrangement of references more uniform.

Point 17: “In Tables 1, 2, and 3, the references are not given at all, are incomplete, or are blanket references. The summarized information cannot be assigned to an origin for the reader”.

Response 17: We did not give references in Table 1, because the classification of cellular stress stages is our idea, not the literature data. However, in the text of the section we have made a justification of this classification and given the relevant references. We have explained this in a note to Table 1.

In Table 2, the references are given in the title (most of them are key reviews); it was difficult to insert them in separate cells, because these references duplicate each other many times, and the table text is already very dense.

In Table 3, the two main references to the fundamental reviews (one of which is ours) are given in the title, and more specific references are given in a note to Table 3 in order to avoid overloading Table 3  with numbers.

Point 18: “Page 25, line 980: Please insert the reference paper for the cited Selye theory”.

Response 18: We have inserted a references to Selye theory.

Round 2

Reviewer 3 Report

The authors have satisfactorily addressed the concerns raised in the original version. The revised version is significantly improved. No further concerns.